# APPROXIMATION ALGORITHMS FOR COMBINATORIAL OPTIMIZATION WITH PREDICTIONS[*]

**Antonios Antoniadis**
University of Twente, Enschede, Netherlands
a.antoniadis@utwente.nl

**Marek Eliáš & Adam Polak & Moritz Venzin**
Bocconi University, Milan, Italy
{marek.elias,adam.polak,moritz.venzin}@unibocconi.it

## ABSTRACT

We initiate a systematic study of utilizing predictions to improve over approximation guarantees of classic algorithms, without increasing the running time. We propose a systematic method for a wide class of optimization problems that ask to select a feasible subset of input items of minimal (or maximal) total weight. This gives simple (near-)linear time algorithms for, e.g., Vertex Cover, Steiner Tree, Min-Weight Perfect Matching, Knapsack, and Clique. Our algorithms produce optimal solutions when provided with perfect predictions and their approximation ratios smoothly degrade with increasing prediction error. With small enough prediction error we achieve approximation guarantees that are beyond reach without predictions in the given time bounds, as exemplified by the NP-hardness and APX-hardness of many of the above problems. Although we show our approach to be optimal for this class of problems as a whole, there is a potential for exploiting specific structural properties of individual problems to obtain improved bounds; we demonstrate this on the Steiner Tree problem.

## 1 INTRODUCTION

Combinatorial optimization studies problems of selecting an optimal solution, with respect to a given cost function, from a discrete set of potential solutions. This framework is ubiquitous, both in theory and in practice, finding application in a vast number of areas, e.g., resource allocation, machine learning, efficient networks, or logistics. However, these problems are typically NP-hard, and hence they cannot be solved to optimality in polynomial time (unless P equals NP). It is thus necessary to develop efficient algorithms even if this comes at the cost of relaxing the optimality requirement. This trade-off between optimality and tractability is the main paradigm of the (classic) theory of approximation algorithms, see, e.g., (Williamson and Shmoys, 2011).

Despite the success of the classic analysis of approximation algorithms, its distinction between tractable and intractable problems is often too crude. Indeed, for modern big data applications, a running time polynomial in the input size can hardly be considered efficient, as we often require algorithms with a running time that is a small polynomial or even linear in the input size. This more fine-grained point of view on algorithm design has received considerable attention in recent years. Many combinatorial optimization problems (e.g., Vertex Cover, Steiner Tree, or Knapsack) admit simple and fast (near-)linear time constant-factor approximation algorithms (Bar-Yehuda and Even, 1981; Mehlhorn, 1988; Dantzig, 1957), but improving upon their approximation guarantees often seems to require significantly more running time (sometimes even exponentially, as is the case for Vertex Cover (Khot and Regev, 2008)). However, it is important to note that such limitations are based on worst-case analysis, and do not necessarily represent the difficulty of a typical instance.

It is natural to expect that instances arising in practice can be modelled as coming from a fixed distribution, making them amenable to machine-learning techniques. Using historic input data and

---

[*]The full version of the paper is available at arxiv.org/abs/2411.16600.

some (possibly computationally expensive) training, we can hope to guide an efficient algorithm with poor (worst-case) approximation guarantee to obtain near-optimal solutions.

## 1.1 OUR CONTRIBUTION

In this paper, we initiate a systematic study of utilizing predictions to improve over the approximation guarantees of classic algorithms without increasing the running time. This approach fits the current line of research on utilizing predictions to improve performance of algorithms, started by the seminal papers of Kraska et al. (2018) and Lykouris and Vassilvitskii (2021). We focus on a broad class of optimization problems, which we call *selection problems*, and which captures many fundamental problems in combinatorial optimization, e.g., Set Cover, Travelling Salesperson Problem (TSP), Steiner Tree, or Knapsack.

**Definition 1** (Selection problem). *We are given $n$ items, numbered with integers from $[n] := \{1, 2, \ldots, n\}$, with non-negative weights $w \colon [n] \to \mathbb{R}_{\geqslant 0}$ and an implicit collection $\mathcal{X} \subseteq 2^{[n]}$ of feasible subsets of items. Our task is to find a feasible solution $X \in \mathcal{X}$ minimizing (or maximizing) its total weight $w(X) := \sum_{i \in X} w(i)$.*

Note that the complexity of this task comes from the size of $\mathcal{X}$, which is usually exponential in $n$.

Our work considers the *offline* setting, where the whole input is available to the algorithm from the start. This is in contrast to much of the to-date research on learning-augmented algorithms, which focuses on *online* problems, where the input is not known to the algorithm in advance and the predictions are used to decrease uncertainty about the future (see the survey of Mitzenmacher and Vassilvitskii (2020)). So far, offline problems were studied mostly in the *warm-start* setting, where predictions in the form of solutions to past instances are used to speed up exact algorithms (see Section 1.4 on related work). The focus of such works is on the dependence between the running time and the quality of the predictions.

Our target is to maintain a superb running time in all situations, and we study the dependence of approximation ratio on the prediction quality. Having access to past instances, one can hope to learn which items are likely to be part of an optimal solution and provide the set of such items to the algorithm as a prediction. Utilizing such prediction comes with two main challenges.

*(1) The predicted set of items may be infeasible.* Ensuring feasibility of solutions produced by deep-learning models is a challenging problem, and enforcing even simple combinatorial constraints requires very complex neural architectures (Bengio et al., 2021). In order to utilize potentially infeasible predictions effectively, we need algorithms that can transform the prediction into a feasible solution without discarding the correct parts of the prediction. We carefully design a *black-box* mechanism that turns *any* approximation algorithm into a learning-augmented algorithm (utilizing possibly infeasible predictions) whose approximation ratio smoothly deteriorates with increasing prediction error. Our approach gives (near-)linear time algorithms for many problems, e.g., Vertex Cover, Steiner Tree, Min-Weight Perfect Matching, Knapsack, and Clique.

*(2) The predicted set of items may contain costly mispredictions.* When given a mostly correct prediction, it is certainly possible to detect a single mispredicted item with a huge weight, say, larger than the total weight of the optimal solution. Our goal is to push this idea to its limits by detecting mispredictions which are as small as possible. We study this challenge on the case of the Steiner Tree problem. We develop an algorithm that can detect and eliminate mispredictions of cost larger than the cost of a single connection between a pair of terminals in an optimal solution.

## 1.2 OUR RESULTS IN MORE DETAIL

Before stating our results formally, we have to define two crucial quantities: the approximation ratio and the prediction error.

**Approximation ratio.** We measure the quality of solutions produced by approximation algorithms using the standard notion of *approximation ratio*. For minimization problems, we say that an algorithm is a $\rho$-approximation algorithm if we are always guaranteed that $w(X) \leqslant \rho \cdot w(X^*)$, where $X$ denotes the solution returned by the algorithm, and $X^*$ denotes an optimal solution for the same instance. Analogously, for maximization problems, a $\rho$-approximate algorithm always satisfies

$w(X) \geqslant \rho \cdot w(X^*)$. We refer to $\rho$ as the approximation ratio. For minimization problems we have $\rho \geqslant 1$ and smaller ratios are better; for maximization problems $\rho \in [0, 1]$ and larger ratios are better.

**Prediction and prediction error.** The prediction received by our algorithms is an arbitrary (not necessarily feasible) set of items $\widehat{X} \subseteq [n]$. We say that $\widehat{X}$ *has error* $(\eta^+, \eta^-)$ *with respect to a solution* $X$, if

$$\eta^+ := w(\widehat{X} \setminus X), \quad \text{and} \quad \eta^- := w(X \setminus \widehat{X}),$$

i.e., $\eta^+$ is the total weight of false positives, and $\eta^-$ is the total weight of false negatives. Usually we consider the prediction error with respect to an optimal solution, denoted by $X^*$. We note that such predictions are PAC-learnable (see Section 1.3). In our performance bounds, the prediction error can be always considered with respect to *the closest* optimal solution.

Let us also remark that an alternative setting, where each item $i \in [n]$ is associated with a fractional prediction $p_i \in [0, 1]$ (supposed to denote the a *confidence* that $i$ is part of the optimal solution), is actually equivalent to the setting we consider. Indeed, one can convert such fractional predictions to a set $\hat{X}$ with simple randomized rounding (adding each item $i$ to $\hat{X}$ independently with probability $p_i$) and the expected value of the prediction error stays the same.

**Minimization problems.** Let $\Pi$ be a problem of selecting a feasible solution $X \in \mathcal{X}$ of minimum total weight. In Section 2, we show a black-box approach to turn any off-the-shelf $\rho$-approximation algorithm $A$ for $\Pi$ into a learning-augmented algorithm for $\Pi$ with approximation ratio

$$1 + \frac{\eta^+ + (\rho - 1) \cdot \eta^-}{\text{OPT}}.$$

Here, $(\eta^+, \eta^-)$ denotes the error of the prediction $\widehat{X}$ given to the algorithm with respect to an optimal solution $X^*$ of weight $\text{OPT} := w(X^*)$. The asymptotic running time of the resulting algorithm is the same as that of $A$.

To gain some intuition about the above approximation ratio guarantee, note that for the trivial prediction $\widehat{X} = \emptyset$ we have $\eta^+ = 0$ and $\eta^- = \text{OPT}$, and in turn $1 + (\eta^+ + (\rho - 1) \cdot \eta^-)/\text{OPT} = \rho$, which squarely corresponds to simply running the (prediction-less) algorithm $A$. This shows that the $\rho - 1$ factor in front of $\eta^-$ is necessary.

In Section 2.1 we give several examples of how this black-box approach can be applied in order to obtain (near-)linear time learning-augmented approximation algorithms for some fundamental problems in combinatorial optimization, namely Minimum (and Min-Weight) Vertex Cover, Minimum (and Min-Weight) Steiner Tree, and Min-Weight Perfect Matching (in graphs with edge weights satisfying the triangle inequality).

**Maximization problems.** In Section 3 we give a similar result for maximization problems. Let $\Pi$ be a problem of selecting a set *maximizing* the total weight from the collection of sets of items $\mathcal{X}$. Let $A$ be a $\rho$-approximation algorithm for the corresponding *complementary* problem of selecting a set *minimizing* the total weight over sets of items $Y$ such that $([n] \setminus Y) \in \mathcal{X}$. We construct an algorithm for $\Pi$ with running time asymptotically the same as that of $A$ and with approximation ratio

$$1 - \frac{(\rho - 1) \cdot \eta^+ + \eta^-}{\text{OPT}}$$

given predictions of error $(\eta^+, \eta^-)$ with respect to an optimal solution $X^*$ of weight $\text{OPT} := w(X^*)$.

Even though the notion of the complementary problem may not seem intuitive at first, we note that for many natural problems the complementary problem also happens to be a natural and well studied problem, e.g., Vertex Cover is the complementary problem of Independent Set. In Section 3.1 we elaborate on how to apply our black-box construction to Clique and Knapsack.

**Lower bounds.** In the full version of the paper (Antoniadis et al., 2024), we show that our black-box approach from Sections 2 and 3, despite being simple, cannot be improved for the class of selection problems as a whole. More specifically, regarding minimization problems, we show that for the Vertex Cover problem with predictions, any (polynomial-time) learning-augmented algorithm with an approximation ratio with a better dependence on the prediction error would contradict the Unique

Games Conjecture (UGC), which is a standard assumption in computational complexity but to the best of our knowledge has not been used before in the context of learning-augmented algorithms. Regarding maximization problems, we give a similar UGC-based lower bound for the Clique and Independent Set problems.

**Refined algorithm for Steiner Tree.**  Although our lower bounds are tight for the considered classes of combinatorial optimization problems as a whole, they do not rule out refined upper bounds, e.g., for specific problems or in terms of other (more fine-grained) measures of prediction errors. In Section 4 we propose such a refined algorithm for the Steiner Tree problem. In this problem, a small number of false positives with a large weight can have a large impact on the performance of our generic black-box algorithm from Section 2. Our refined algorithm is guided by a hyperparameter $\alpha$ in order to detect and avoid false positives with high weight. It is based on a 2-approximation algorithm called the *MST heuristic* (Kou et al., 1981; Mehlhorn, 1988). The contribution of false positives to our algorithm's performance guarantee depends only on their number, and not on their weights, and it is capped by the cost of individual connections made by the MST heuristic (without predictions). The final approximation guarantee follows from a careful analysis of how the output of our algorithm converges to the prediction as its hyperparameter $\alpha$ increases. We also show that our analysis of this algorithm is tight.

**Experimental results.**  The full version of the paper (Antoniadis et al., 2024) contains an experimental evaluation[1] of our refined Steiner Tree algorithm on graphs from the PACE 2018 Challenge (Bonnet and Sikora, 2018), using as a benchmark the winning Steiner Tree solver from that challenge (Ruiz et al., 2018). These experiments allow us to better understand the impact of the hyperparameter $\alpha$ on the performance of our algorithm. They also demonstrate that (for a sufficiently concentrated input distribution) we can find near-optimal solutions in time in which conventional algorithms can achieve only rough approximations.

### 1.3   Further remarks on our setting

**Learnability.**  The predictions that our algorithms require are PAC-learnable via the following simple argument. First, since the space of possible predictions is finite and its size is single exponential in $n$, it suffices to perform empirical risk minimization (ERM) on a polynomial number of samples (see, e.g., Polak and Zub (2024, Theorem 5)). Second, ERM for the combined prediction error $\eta^+ + \eta^-$ boils down to taking a coordinate-wise majority vote of solutions to the sampled instances. We use this approach in our experiments in the full version of the paper (Antoniadis et al., 2024).

At the same time, our setting is flexible enough to allow for other methods of generating predictions. For instance, it is not hard to imagine a deep-learning model that assigns to each input element the probability that it belongs to an optimal solution (Joshi et al., 2019; Ahn et al., 2020). Then, a prediction can be obtained by sampling each element with the assigned probability.

We remark that any such learning is likely to be computationally expensive, but this should come at no surprise, because the resulting predictions can then be utilized by our learning-augmented algorithms to "break" known lower bounds. The time saved this way must be spent somewhere else, i.e., during learning. The advantage is that, when we are repeatedly solving similar recurring instances, this costly learning process is performed only once, and the resulting predictions can be (re-)used multiple times.

**Infeasible predictions.**  We stress out that it is an absolutely crucial characteristic of our work that our algorithms accept as predictions sets that are not necessarily feasible solutions.[2] Bengio et al. (2021) argue that ensuring feasibility of predictions significantly increases the complexity of the learning process, with difficulties specific to different types of combinatorial constraints. Accepting infeasible predictions allows for simpler and possibly more versatile learning approaches like the ones outlined in the previous paragraphs.

---

[1]Our source code is available at `github.com/adampolak/steiner-tree-with-predictions`.

[2]It is a common characteristic of learning-augmented algorithms. E.g., in the paper on warm-starting max-flow by Davies et al. (2023) most of the technical insights are in the part of the algorithm that projects any (potentially infeasible) prediction into a feasible solution.

Even when the input changes very slightly, a previously feasible solution may not be feasible anymore, so one should not expect feasibility of predictions based on past data. One such example scenario is given in our experiments on the Steiner Tree problem in the full version of the paper (Antoniadis et al., 2024), where the underlying graph is fixed and the set of terminals changes, hence changing the set of feasible solutions. It might be the case that a part of the solution is recurring while the other part is changing constantly, from instance to instance. Then, a learning algorithm can easily predict the stable part of the solution, but in advance it is hard to extend such a partial solution into a complete feasible solution with reasonable accuracy.

### 1.4 Related work

**Learning-augmented approximation algorithms.** Approximation algorithms with predictions have so far received very little attention and have only been investigated for specific problems and in restricted settings.

Bampis et al. (2024) studied *dense* variants of several hard problems – such as Max Cut, Max $k$-SAT, or $k$-Densest Subgraph – which all are examples of maximization selection problems[3]. In the dense setting, these problems admit polynomial-time approximation schemes. In particular, for any fixed $\epsilon > 0$, there is a $(1 - \epsilon)$-approximation algorithm for Max Cut by Arora et al. (1999) that runs in time $O(n^{1/\epsilon^2})$, i.e., exponential in the precision parameter $1/\epsilon$. Bampis et al. (2024) propose an algorithm that always runs in time $O(n^{3.5})$ and achieves approximation ratio depending on the quality of the received prediction. An interesting feature of their work is that the algorithms in the dense setting need only a small sample of the predicted solution to achieve their guarantees. In particular, they only need to know $O(\text{poly}(\log n/\epsilon))$ bits of information about items belonging or not belonging to the optimal solution to achieve an approximation ratio of the form $1 - \epsilon - f(\eta)$, where $\eta$ denotes the prediction error.

Ergun et al. (2022) and Gamlath et al. (2022) independently proposed a linear time algorithm for $k$-Means Clustering that receives labels of the input points as predictions. This problem can be formulated as a minimization selection problem only in the special case where the number of potential centers is bounded (e.g., if the potential centers are the input points themselves). If the prediction has per-cluster label error rate at most $\alpha$ with respect to some $(1 + \alpha)$-approximate solution, their algorithm achieves an approximation ratio of $1 + O(\alpha)$. The algorithm uses techniques from robust statistics to identify outliers in the predicted labeling. Nguyen et al. (2023) slightly improve upon this guarantee in their follow-up work.

There are also two very recent works, on the Max Cut (Cohen-Addad et al., 2024) and Independent Set (Braverman et al., 2024) problems, in the setting with $\epsilon$-*accurate predictions*. In this setting, each input item comes with a label (indicating whether the item belongs to a fixed optimal solution or not), which is correct with probability $1/2 + \epsilon$, *independently* from other labels. The two papers show how to breach the respective approximation ratio barriers of $0.878$ and $n^{1-o(1)}$, for any $\epsilon > 0$. Note that our results assume that the incorrect parts of the prediction are selected adversarially rather than randomly.

**Other learning-augmented algorithms.** Another related line of work is that on exact offline algorithms with *warm start*. In this setting, the algorithm, in contrast to starting its computation from scratch, has access to predictions or information (e.g., a partial solution) that allow it to start computing from a "more advanced" state, leading to an improvement in the running time. Such results include Kraska et al. (2018) on binary search, Dinitz et al. (2021); Chen et al. (2022) on Bipartite Matching, Feijen and Schäfer (2021); Lattanzi et al. (2023) on Shortest Path, Polak and Zub (2024) on Max Flow, and Bai and Coester (2023) on Sorting. There are also works considering multiple predictions (Dinitz et al., 2022; Sakaue and Oki, 2022) and a work by Tang et al. (2020) on using reinforcement learning to improve the cutting plane heuristic for Integer Programming.

Since the seminal papers by Kraska et al. (2018) and Lykouris and Vassilvitskii (2021), which initiated the study of learning-augmented algorithms in the modern sense, many online computational problems were considered. There are papers on, e.g., ski rental (Purohit et al., 2018), secretary problem (Dütting et al., 2024), energy efficient scheduling (Bamas et al., 2020), online page migration

---

[3]E.g., for Max Cut, $\mathcal{X} \subseteq 2^E$ contains all sets of edges that correspond to a cut in the input graph.

(Indyk et al., 2022), online TSP (Bernardini et al., 2022), and flow-time scheduling (Azar et al., 2021; 2022). Further related works can be found on the website by Lindermayr and Megow (2022).

## 2 MINIMIZATION PROBLEMS

Our algorithm for minimization selection problems with linear objective receives a prediction $\widehat{X} \subseteq [n]$ which may not be feasible. It changes the weight of each item in $\widehat{X}$ to $0$ and runs a conventional $\rho$-approximation algorithm on the problem with those modified weights. This way, we obtain a solution which is always feasible and its quality is described by the following theorem. In the full version of the paper (Antoniadis et al., 2024), we show that this simple approach already matches a lower bound based on the UGC.

**Theorem 1.** *Let $\Pi$ be a minimization selection problem, and let $A$ be a $\rho$-approximation algorithm for $\Pi$ running in time $T(n)$. Then, there exists an $O(T(n))$-time learning-augmented approximation algorithm for $\Pi$ with the following guarantee: Upon receiving a (not necessarily feasible) predicted solution $\widehat{X} \subseteq [n]$, it outputs a solution $X$ such that, for any feasible solution $X' \in \mathcal{X}$, we have $w(X) \leqslant w(X') + \eta^+ + (\rho - 1) \cdot \eta^-$, where $(\eta^+, \eta^-)$ is the error of $\widehat{X}$ with respect to $X'$.*

We remark that if $X' = X^*$ is an optimal solution with objective value OPT, the preceding bound implies that our algorithm's approximation ratio is at most

$$1 + \frac{\eta^+ + (\rho - 1) \cdot \eta^-}{\text{OPT}}.$$

*Proof.* The algorithm works as follows: Set

$$\bar{w}(i) = \begin{cases} 0, & \text{if } i \in \widehat{X} \\ w(i), & \text{otherwise} \end{cases} \quad \text{for } i = 1, 2, \ldots, n.$$

Run algorithm $A$ with weight function $\bar{w}$ and return $X$, the solution returned by the algorithm.

We claim that $w(X) \geqslant w(X') + \eta^+ + (\rho - 1)\eta^-$. Since $A$ is a $\rho$-approximation algorithm and $X'$ is a feasible solution for $\bar{w}_1, \bar{w}_2, \ldots, \bar{w}_n$, it holds that

$$w(X \setminus \widehat{X}) = \bar{w}(X) \leqslant \rho \cdot \bar{w}(X') = \rho \cdot w(X' \setminus \widehat{X}).$$

Then,

$$\begin{aligned} w(X) &= w(X \cap \widehat{X}) + w(X \setminus \widehat{X}) \\ &\leqslant w(X \cap \widehat{X}) + \rho \cdot w(X' \setminus \widehat{X}) \\ &= w(X \cap \widehat{X}) + w(X' \setminus \widehat{X}) + (\rho - 1) \cdot w(X' \setminus \widehat{X}) \\ &\leqslant w(\widehat{X}) + w(X' \setminus \widehat{X}) + (\rho - 1) \cdot w(X' \setminus \widehat{X}). \end{aligned}$$

Note that

$$w(\widehat{X}) + w(X' \setminus \widehat{X}) = w(\widehat{X} \cup X') = w(X') + w(\widehat{X} \setminus X').$$

Thus we have,

$$w(X) \leqslant w(X') + w(\widehat{X} \setminus X') + (\rho - 1) \cdot w(X' \setminus \widehat{X}) = w(X') + \eta^+ + (\rho - 1)\eta^-,$$

where $(\eta^+, \eta^-)$ is the error of $\widehat{X}$ with respect to $X'$. If $X' = X^*$ is an optimal solution, we have

$$w(X) \leqslant \text{OPT} + \eta^+ + (\rho - 1) \cdot \eta^- = \left( 1 + \frac{\eta^+ + (\rho - 1) \cdot \eta^-}{\text{OPT}} \right) \cdot \text{OPT}. \qquad \square$$

In principle, the false-positive prediction error $\eta^+$ can be unbounded in terms of OPT, and therefore the approximation ratio of the algorithm of Theorem 3 cannot be bounded by any constant. That is, using the terminology of learning-augmented algorithms, the algorithm is not *robust*. However, as is the case for any offline algorithm, it can be robustified without increasing the asymptotic running time by simply running $A$ in parallel and returning the better of the two solutions.

**Corollary 2.** *Under the same assumptions as in Theorem 1 there exists a learning augmented algorithm for $\Pi$ running in time $O(T(n))$ with approximation ratio*

$$\min \left\{ 1 + \frac{\eta^+ + (\rho - 1) \cdot \eta^-}{\text{OPT}}, \rho \right\}.$$

## 2.1 EXAMPLE APPLICATIONS

**Minimum (and Min-Weight) Vertex Cover.** In an undirected graph $G = (V, E)$, a *vertex cover* is a set $X \subseteq V$ such that every edge $e = (u, v) \in E$ has $u \in X$ or $v \in X$. The Min-Weight Vertex Cover problem asks, given a graph $G = (V, E)$ with vertex weights $w : V \to \mathbb{R}_{\geqslant 0}$, to find a vertex cover $X \subseteq V$ of the minimum total weight $w(X)$, and the Minimum Vertex Cover problem is the special case with unit weights $\forall_{u \in V} w(u) = 1$, i.e., it asks for a vertex cover of minimum cardinality.

Already the unweighted variant is a very hard problem: Minimum Vertex Cover is included in Karp's seminal list of 21 NP-hard problems (Karp, 1972). Furthermore, under the UGC it is NP-hard to approximate Minimum Vertex Cover with any better than 2 multiplicative factor (Khot and Regev, 2008). This lower bound matches a folklore 2-approximation algorithm, which runs in linear time. Bar-Yehuda and Even (1981) show that also Min-Weight Vertex Cover admits a linear-time 2-approximation. Therefore, we can directly apply Theorem 1 and get a linear-time learning-augmented algorithm for Min-Weight Vertex Cover with the approximation ratio $1 + (\eta^+ + \eta^-)/\text{OPT}$.

**Minimum (and Min-Weight) Steiner Tree.** Given an undirected graph $G = (V, E)$ and a subset $T \subseteq V$ of the vertices referred to as *terminals*, the Minimum Steiner Tree problem asks for a set of edges $X \subseteq E$ of minimum cardinality such that all terminals in $T$ belong to the same connected component of $(V, X)$. In the Min-Weight Steiner Tree problem the input also contains edge weights $w : E \to \mathbb{R}_{\geqslant 0}$ and the goal is to find such set $X$ minimizing the total weight $w(X)$.

Minimum Steiner Tree is also among Karp's 21 NP-hard problems (Karp, 1972). A folklore algorithm, the so-called *minimum spanning tree heuristic* yields 2-approximation algorithm, also for the Min-Weight Steiner Tree problem, and Mehlhorn (1988) shows how to implement it in (near-)linear time $O(|E| + |V| \log |V|)$. A long line of work contributed many (polynomial-time) better-than-2-approximation algorithms, with the current best approximation factor 1.39 given by Byrka et al. (2013), but none of these algorithms runs in (near-)linear time and for many of them the running time is an unspecified polynomial with a huge exponent. The inapproximability lower bound is $96/95$ (Chlebík and Chlebíková, 2008), leaving a big gap open.

Our Theorem 1 together with Mehlhorn's algorithm gives a linear time learning-augmented Min-Weight Steiner Tree algorithm with approximation factor $1 + (\eta^+ + \eta^-)/\text{OPT}$. For sufficiently accurate predictions, it gives better and faster approximation than the best conventional algorithms. In Section 4 we show how to exploit specific structural properties of the problem in order to obtain an even better algorithm.

**Min-Weight Perfect Matching.** For an undirected graph $G = (V, E)$ with edge weights $w : E \to \mathbb{R}_{\geqslant 0}$, the Min-Weight Perfect Matching problem asks to find a set $X \subseteq E$ of exactly $|X| = |V|/2$ edges such that each vertex $u \in V$ is an endpoint of exactly one of these edges and their total weight $w(X)$ is minimized. Unlike the previous two examples, this problem belongs to the class P and optimal solutions can be found with a fairly complicated exact algorithm that runs in $O(|V||E|)$ time (Edmonds, 1965; Gabow, 1990). For the special case of edge weights satisfying the triangle inequality[4] Goemans and Williamson (1995) give a linear-time 2-approximation algorithm. It is an open problem whether a better-than-2-approximation faster than $O(|V||E|)$ is possible for this special case. We show how to achieve it assuming sufficiently accurate predictions.

This time we need to work more than in the previous two examples. That is because the problem we tackle is not a selection problem – because of the triangle inequality not every weight function constitutes a correct input. If we tried to apply Theorem 1 directly we would run into the issue that decreasing the weights of predicted edges to zero may violate the triangle inequality.

Let us start by defining the so-called $V$-join problem: Given an edge-weighted graph find a min-weight set of edges that has odd degree on all the vertices of $V$. Goemans and Williamson (1995) give a 2-approximation algorithm for the $V$-join problem. They also show how to short-cut a $\rho$-approximate solution to the $V$-join problem to get a $\rho$-approximate solution to the Min-Weight Perfect Matching, provided that the edge weights satisfy the triangle inequality. Their algorithm runs in near-linear time.

---

[4]This special case is a subproblem in the famous 1.5-approximation TSP algorithm by Christofides (2022).

Our learning-augmented algorithm for Min-Weight Perfect Matching works in two steps. First, it uses Theorem 1 with the algorithm of Goemans and Williamson (1995) to find a solution to the $V$-join problem with approximation ratio at most $1 + (\eta^+ + \eta^-)/\mathrm{OPT}$. Then, provided that the original graph satisfies the triangle inequality, it transforms the $V$-join solution into a perfect matching with the same approximation ratio, using the short-cutting procedure of Goemans and Williamson (1995).

## 3 MAXIMIZATION PROBLEMS

Our algorithm for maximization selection problems with linear objective receives a prediction $\widehat{X} \subseteq [n]$ which may not be feasible. It changes the weight of each item in $[n] \setminus \widehat{X}$ to 0 and runs the $\rho$-approximation algorithm for the *complementary* problem to find a set $Y$. This way, it obtains a solution $X = [n] \setminus Y$ that is guaranteed to be feasible. Its quality is given by the following theorem. In the full version of the paper (Antoniadis et al., 2024) we show that this simple approach already matches a lower bound based on UGC.

**Theorem 3.** *Let $\Pi$ be some maximization selection problem. Let $A$ be a $T(n)$-time $\rho$-approximation algorithm for the following complementary problem: Find a subset of items $Y \subseteq [n]$ minimizing $w(Y)$ such that $([n] \setminus Y) \in \mathcal{X}$. Then, there exists an $O(T(n))$ time learning-augmented approximation algorithm for $\Pi$ with the following performance. Receiving a (not necessarily feasible) predicted solution $\widehat{X} \subseteq [n]$, it outputs a solution $X$ such that, for any feasible solution $X' \in \mathcal{X}$, we have $w(X) \geqslant w(X') - (\rho - 1)\eta^+ - \eta^-$, where $(\eta^+, \eta^-)$ is the error of $\widehat{X}$ with respect to $X'$.*

*If $X' = X^*$ is an optimal solution with objective value $\mathrm{OPT}$, the preceding bound implies that our algorithm's approximation ratio is at most*

$$1 - \frac{(\rho - 1) \cdot \eta^+ + \eta^-}{\mathrm{OPT}}.$$

*Proof.* The algorithm works as follows: Set

$$\bar{w}(i) = \begin{cases} w(i), & \text{if } i \in \widehat{X} \\ 0, & \text{otherwise} \end{cases} \quad \text{for } i \in [n].$$

Run algorithm $A$ with weight function $\bar{w}$ and let $Y$ denote the solution returned by the algorithm. Return $X = [n] \setminus Y$.

By the definition of the complementary problem, we have that $X = [n] \setminus Y \in \mathcal{X}$ is a feasible solution to $\Pi$. It remains to lower bound the weight of that solution in terms of $w(X')$. Note that

$$w(\widehat{X}) = w(X') + w(\widehat{X} \setminus X') - w(X' \setminus \widehat{X}).$$

Moreover, $[n] \setminus X'$ is a feasible solution to the complementary problem with weights $\bar{w}$, because $X' \in \mathcal{X}$. Hence, since $A$ is a $\rho$-approximation algorithm, we get

$$\bar{w}(Y) \leqslant \rho \cdot \bar{w}\big([n] \setminus X'\big) = \rho \cdot w(\widehat{X} \setminus X').$$

Finally, we have

$$\begin{aligned} w(X) = w\big([n] \setminus Y\big) &\geqslant w(\widehat{X}) - \bar{w}(Y) \\ &\geqslant w(X') + w(\widehat{X} \setminus X') - w(X' \setminus \widehat{X}) - \rho w(\widehat{X} \setminus X') \\ &= w(X') - (\rho - 1)\eta^+ - \eta^-, \end{aligned}$$

where $(\eta^+, \eta^-)$ is the error of $\widehat{X}$ with respect to $X'$. If $X'$ is an optimal solution, we have

$$w(X) \geqslant \mathrm{OPT} - (\rho - 1)\eta^+ - \eta^- = \left(1 - \frac{(\rho - 1) \cdot \eta^+ + \eta^-}{\mathrm{OPT}}\right) \cdot \mathrm{OPT}. \qquad \square$$

Similarly to Theorem 1, the algorithm implied by Theorem 3 can be robustified by running it in parallel with a conventional approximation algorithm $A'$ for problem $\Pi$.

**Corollary 4.** *Under the same assumptions as in Theorem 3 there exists a learning augmented algorithm for $\Pi$ running in time $O(T(n))$ with approximation ratio*

$$\max\left\{1 - \frac{(\rho - 1)\eta^+ + \eta^-}{\mathrm{OPT}}, \rho'\right\},$$

*where $\rho'$ is the approximation ratio of the best known algorithm for $\Pi$ running in time $O(T(n))$.*

### 3.1 Example applications

**Maximum (and Max-Weight) Clique (and Independent Set).** Given an undirected graph $G = (V, E)$, an *independent set* (also called *stable set*) is a subset of vertices $X \subseteq V$, such that no two vertices in $X$ share an edge. The Maximum Independent Set problem asks for an independent set of largest cardinality. In the Max-Weight Independent Set problem the input also includes vertex weights $w : V \to \mathbb{R}_{\geqslant 0}$ and the goal is to find an independent set $X$ maximizing the total weight $w(X)$. In the complement graph $G' = \left(V, \binom{V}{2} \setminus E\right)$, these problems are equivalent to the Maximum (and Max-Weight) Clique problems, which ask for the largest cardinality (weight) complete subgraph $K_\ell \subseteq G$.

Maximum Clique (and Independent Set) is not only NP-hard to solve exactly (Karp, 1972), but it is also NP-hard to approximate within any factor better than $n^{1-\epsilon}$, for any $\epsilon > 0$ (Håstad, 1999). Quite conveniently for us, the complementary problem to Max-Weight Independent Set is Min-Weight Vertex Cover, which can be 2-approximated in linear time (Bar-Yehuda and Even, 1981). Thus, applying Theorem 3, we get a linear-time learning-augmented algorithm for Min-Weight Independent Set (and Clique) with approximation ratio $1 - (\eta^+ + \eta^-)/\text{OPT}$, which is in striking contrast to the aforementioned impossibility of any nontrivial approximation ratio for conventional algorithms.

**Knapsack.** Given the knapsack capacity $c$ and $n$ items, the $i$-th of which has size $s_i$ and is worth $w_i$, the Knapsack problem asks to find a subset of items of total size at most $c$ that maximizes the total worth. Knapsack is another example from Karp's list of NP-hard problems (Karp, 1972), but it admits approximation schemes. In particular, it can be approximated to within factor $(1 - \epsilon)$ in time $\tilde{O}(n + \epsilon^{-2})$ (Chen et al., 2024; Mao, 2024). Unfortunately, these asymptotically optimal algorithms are based on structural results from additive combinatorics that make the constants hidden in the asymptotic notation enormous and render the algorithms themselves impractical. Perhaps a more practical approach is the $O(n \log(\epsilon^{-1}) + \epsilon^{-2.5})$-time algorithm by Chan (2018). Still, if our goal is to solve with high accuracy (i.e., small $\epsilon$) many similar Knapsack instances, it might be a useful strategy to use Chan's algorithm only on a small fraction of those instances, learn from the obtained solutions a predicted solution, and feed it to a much faster learning-augmented algorithm in order to solve the remaining majority of instances.

In the following lemma (proved in the full version of the paper (Antoniadis et al., 2024)) we show that the problem complementary to Knapsack admits an $O(n \log n)$-time 2-approximation algorithm. Theorem 3 then implies the existence of an $O(n \log n)$-time learning-augmented algorithm for Knapsack with approximation ratio $1 - (\eta^+ + \eta^-)/\text{OPT}$.

**Lemma 1.** *There is an $O(n \log n)$-time $2$-approximation algorithm for the following problem: Given $n$ items, the $i$-th of which has size $s_i$ and is worth $w_i$, and the target $t$, find a subset of items of total size at least $t$ that minimizes the total worth.*

## 4 Refined bounds for Steiner Tree

We describe a refined algorithm for Steiner Tree based on a 2-approximation algorithm by Mehlhorn (1988). Our careful analysis describes how it detects and avoids false positives with high weight. Our algorithm uses a parameter $\alpha \geqslant 1$ to adapt its treatment of the prediction. With $\alpha = 1$ its behavior copies Mehlhorn's algorithm, and $\alpha$ approaching infinity corresponds to the algorithm in Theorem 1. However, a different choice of $\alpha$ may give better results. We illustrate this by an example and show how to achieve performance close to the best choice of $\alpha$ with only a constant factor increase in running time in the full version of the paper (Antoniadis et al., 2024).

The algorithm receives as input a graph $G = (V, E)$, set $T \subseteq V$ of $k$ terminals, a weight function $w : E \to \mathbb{R}_{\geqslant 0}$, and a set $\widehat{X} \subseteq E$ of predicted edges. First, it scales down the weight of edges in $\widehat{X}$ by dividing them by the parameter $\alpha$. Then, it uses the algorithm of Mehlhorn (1988) to compute a minimum spanning tree $\text{MST}_\alpha$ of the metric closure of the terminals with respect to the scaled edge weights. In the end, the algorithm outputs the union of edges contained in paths corresponding to the connections in $\text{MST}_\alpha$. The algorithm is summarized in Algorithm 1.

**Proposition 1** (Mehlhorn (1988)). *The minimum spanning tree of the metric closure $\left(T, \binom{T}{2}\right)$ of the graph $G$ can be computed in time near-linear in the size of $G$.*

---

**Algorithm 1:** Steiner tree with predictions

---

**Parameter:** $\alpha \geqslant 1$;
**foreach** $e \in E$ **do**
    **if** $e \in \widehat{X}$ **then** $w_\alpha(e) := w(e)/\alpha$;
    **else** $w_\alpha(e) := w(e)$;

Compute $\mathrm{MST}_\alpha$ of the metric closure $\mathbf{G} = (T, \binom{T}{2})$ of $G$ with weights $w_\alpha$ using Proposition 1;
$X := \emptyset$;
**foreach** edge $e = \{t_1, t_2\}$ **in** $\mathrm{MST}_\alpha$ **do**
    Choose $p(e) \subseteq E$ the cheapest path in $G$ from $t_1$ to $t_2$ with respect to $w_\alpha$;
    $X := X \cup p(e)$;
**return** $X$;

---

## 4.1 Analysis

Recall that $G = (V, E)$ denotes the input graph, and let $\mathbf{G} = (T, \binom{T}{2})$ denote the complete graph on the terminals $T \subseteq V$. For any weight function $w : E \to \mathbb{R}_{\geqslant 0}$, consider the shortest path metric on terminals induced by $w$, i.e., for an edge $e$ in $\mathbf{G}$ between $t_1$ and $t_2$, its cost $\mathbf{c}(e)$ is equal to the length of the shortest path $p(e)$, with respect to $w$, from $t_1$ to $t_2$ in $G$. There is a natural correspondence between an edge $e$ (the cost $\mathbf{c}(e)$) in $\mathbf{G}$ and the set of edges $p(e) \subseteq E$ in $G$ (weight $w(p(e))$). Let MST denote a minimum spanning tree on $\mathbf{G}$. Our algorithm satisfies the following performance bound.

**Theorem 5.** *Consider a graph $G = (V, E)$ with edge weights $w : E \to \mathbb{R}_{\geqslant 0}$ and a set $T \subseteq V$ of $k$ terminals. Let $X' \subseteq E$ be any Steiner tree on $G$ and $S$ be a set of $\min\{k - 1, |\widehat{X} \setminus X'|\}$ edges with the highest cost contained in MST. Our refined algorithm outputs a Steiner tree $X$ of total weight*

$$w(X) \leqslant \left(1 + \frac{1}{\alpha}\right) w(X') + \left(1 - \frac{1}{\alpha}\right) \eta^- + \min\left\{\eta^+, (\alpha - 1) \cdot \sum_{e \in S} \mathbf{c}(e)\right\}, \qquad (1)$$

*where $(\eta^+, \eta^-)$ is the prediction error of $\widehat{X}$ with respect to $X'$.*

With $X' = X^*$ being an optimal solution of cost OPT and $\alpha = 1$, the bound in (1) is equal to 2OPT, which corresponds to the conventional algorithm which ignores the predictions. With $\alpha$ approaching infinity, its limit is $\mathrm{OPT} + \eta^+ + \eta^-$, matching the result from Theorem 1. However, it can be much better than both in case of $\widehat{X} \setminus X^*$ containing a small number of edges of very high weight. Compared to Theorem 1, $\eta^+$ in (1) is capped by $(\alpha - 1) \sum_{e \in S} \mathbf{c}(e)$ where $\sum_{e \in S} \mathbf{c}(e) \leqslant \mathbf{c}(\mathrm{MST}) \leqslant 2w(X^*)$, since Mehlhorn's algorithm is a 2-approximation algorithm for Minimum Steiner Tree. Since we always have $\eta^- \leqslant w(X^*)$, (1) shows that Algorithm 1 is a $2\alpha$-approximation algorithm regardless of the prediction error.

We defer the rest of the analysis to the full version of the paper (Antoniadis et al., 2024).

## 5 Discussion

We initiated the study of algorithms with predictions with a focus on improving over the approximation guarantees of classic algorithms without increasing the running time. This paper focused on the wide and important class of selection problems, but it would be interesting to investigate whether similar results can be obtained for central combinatorial optimization problems that do not belong to this class, e.g., clustering and scheduling problems or problems with non-linear (e.g., submodular) objectives.

We demonstrated, with the example of the Steiner Tree problem, that refined algorithms with improved guarantees are possible for specific problems. A second, implicit, advantage of our refined Steiner Tree algorithm is that its actual performance could be bounded in terms of a quantity directly related to the optimal cost, thus avoiding the additional robustification step. An interesting direction for further research would be to identify other problems satisfying these two properties.

## ACKNOWLEDGEMENTS

Marek Eliáš and Moritz Venzin were supported by the European Union - NextGenerationEU, in the framework of the FAIR - Future Artificial Intelligence Research project (FAIR PE00000013 – CUP B43C22000800006). The views and opinions expressed are solely those of the authors and do not necessarily reflect those of the European Union, nor can the European Union be held responsible for them.

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
