---

**Algorithm 1:** Approximation algorithm for the problem complementary to Knapsack

---

Sort items in the increasing order of $\frac{w_i}{s_i}$;

$X := \emptyset$;

$\mathcal{C} := \emptyset$;

**for** $i = 1, \ldots, n$ **do**

    **if** $s(X) + s_i < t$ **then**

        $\big\lvert$  $X := X \cup \{i\}$;

    **else**

        $\big\lfloor$  $\mathcal{C} := \mathcal{C} \cup \{X\}$;

**return** $\arg\min\{w(Y) \mid Y \in \mathcal{C}\}$;

---

*Proof.* The algorithm (see Algorithm 1) maintains an initially empty partial solution $X$, and keeps adding items to it in a greedy manner in the increasing order of the worth-to-size ratio $\frac{w_i}{s_i}$. Whenever adding the next item $i$ to $X$ would make the total size of $X$ meet or exceed the target size $t$, the algorithm skips adding that item to $X$ and instead adds $X \cup \{i\}$ to the set of candidate solutions $\mathcal{C}$. At the end, the algorithm returns the cheapest solution out of the candidate solutions $\mathcal{C}$.

To show that this is indeed a $2$-approximation algorithm, let $\text{OPT} \subseteq [n]$ denote an optimal solution, and for every $i \in \{0, 1, \ldots, n\}$ let $X_i$ denote the set $X$ after the algorithm considered the first $i$ items. It must be that $\text{OPT} \setminus X_n \neq \emptyset$, because $s(X_n) < t \leqslant s(\text{OPT})$. Consider $i = \min(\text{OPT} \setminus X_n)$, recalling that the items are numbered in an increasing order of the worth-to-size ratios. Since $i \notin X_i \subseteq X_n$, it must hold that $s(X_{i-1}) + s_i \geqslant t$, and thus $X_{i-1} \cup \{i\} \in \mathcal{C}$. By the minimality of $i$, we know that $\text{OPT} \cap \{1, \ldots, i-1\} \subseteq X_{i-1}$, and thus each item in $X_{i-1} \setminus \text{OPT}$ has smaller worth-to-size ratio than any item in $\text{OPT} \setminus X_{i-1}$. This, together with the fact that $s(X_{i-1}) < t \leqslant s(\text{OPT})$ and hence also $s(X_{i-1} \setminus \text{OPT}) < s(\text{OPT} \setminus X_{i-1})$, implies that $w(X_{i-1} \setminus \text{OPT}) < w(\text{OPT} \setminus X_{i-1})$ and as a consequence $w(X_{i-1}) < w(\text{OPT})$. Clearly also $w_i \leqslant w(\text{OPT})$, so $\min\{w(Y) \mid Y \in \mathcal{C}\} \leqslant w(X_{i-1} \cup \{i\}) = w(X_{i-1}) + w_i < 2 \cdot \text{OPT}$. $\qquad\square$

## 4 Refined bounds for Steiner Tree

We describe a refined algorithm for Steiner Tree based on a $2$-approximation algorithm by Mehlhorn (1988). Our careful analysis describes how it detects and avoids false positives with high weight. Our algorithm uses a parameter $\alpha \geqslant 1$ to adapt its treatment of the prediction. With $\alpha = 1$ its behavior copies Mehlhorn's algorithm, and $\alpha$ approaching infinity corresponds to the algorithm in Theorem 2. However, a different choice of $\alpha$ may give better results. We illustrate this by an example and show how to achieve performance close to the best choice of $\alpha$ with only a constant factor increase in running time in Section 4.2.

The algorithm receives as input a graph $G = (V, E)$, set $T \subseteq V$ of $k$ terminals, a weight function $w : E \to \mathbb{R}_{\geqslant 0}$, and a set $\widehat{X} \subseteq E$ of predicted edges. First, it scales down the weight of edges in $\widehat{X}$ by dividing them by the parameter $\alpha$. Then, it uses the algorithm of Mehlhorn (1988) to compute a minimum spanning tree $\text{MST}_\alpha$ of the metric closure of the terminals with respect to the scaled edge weights. In the end, the algorithm outputs the union of edges contained in paths corresponding to the connections in $\text{MST}_\alpha$. The algorithm is summarized in Algorithm 2.

**Proposition 6** (Mehlhorn (1988)). *The minimum spanning tree of the metric closure $\left(T, \binom{T}{2}\right)$ of the graph $G$ can be computed in time near-linear in the size of $G$.*

---

**Algorithm 2:** Steiner tree with predictions

---

**Parameter:** $\alpha \geqslant 1$;

**foreach** $e \in E$ **do**
    **if** $e \in \widehat{X}$ **then** $w_\alpha(e) := w(e)/\alpha$;
    **else** $w_\alpha(e) := w(e)$;

Compute $\mathrm{MST}_\alpha$ of the metric closure $\mathbf{G} = (T, \binom{T}{2})$ of $G$ with weights $w_\alpha$ using Proposition 6;
$X := \emptyset$;

**foreach** edge $e = \{t_1, t_2\}$ **in** $\mathrm{MST}_\alpha$ **do**
    Choose $p(e) \subseteq E$ the cheapest path in $G$ from $t_1$ to $t_2$ with respect to $w_\alpha$;
    $X := X \cup p(e)$;

**return** $X$;

---

## 4.1 Analysis

Recall that $G = (V, E)$ denotes the input graph, and let $\mathbf{G} = (T, \binom{T}{2})$ denote the complete graph on the terminals $T \subseteq V$. For any weight function $w : E \to \mathbb{R}_{\geqslant 0}$, consider the shortest path metric on terminals induced by $w$, i.e., for an edge $e$ in $\mathbf{G}$ between $t_1$ and $t_2$, its cost $\mathbf{c}(e)$ is equal to the length of the shortest path $p(e)$, with respect to $w$, from $t_1$ to $t_2$ in $G$. There is a natural correspondence between an edge $e$ (the cost $\mathbf{c}(e)$) in $\mathbf{G}$ and the set of edges $p(e) \subseteq E$ in $G$ (weight $w(p(e))$). Let $\mathrm{MST}$ denote a minimum spanning tree on $\mathbf{G}$. Our algorithm satisfies the following performance bound.

**Theorem 7.** *Consider a graph $G = (V, E)$ with edge weights $w : E \to \mathbb{R}_{\geqslant 0}$ and a set $T \subseteq V$ of $k$ terminals. Let $X' \subseteq E$ be any Steiner tree on $G$ and $S$ be a set of $\min\{k - 1, |\widehat{X} \setminus X'|\}$ edges with the highest cost contained in $\mathrm{MST}$. Our refined algorithm outputs a Steiner tree $X$ of total weight*

$$w(X) \leqslant \left(1 + \frac{1}{\alpha}\right) w(X') + \left(1 - \frac{1}{\alpha}\right) \eta^- + \min\left\{\eta^+, (\alpha - 1) \cdot \sum_{e \in S} \mathbf{c}(e)\right\}, \qquad (1)$$

*where $(\eta^+, \eta^-)$ is the prediction error of $\widehat{X}$ with respect to $X'$.*

A crucial property of Algorithm 2 is that it never buys an edge with weight more than $\alpha$ times larger than some connection in $\mathrm{MST}$. The sum in (1) needs to be over connections in $\mathrm{MST}$ instead of individual edges in $X'$, since $X'$ may consist of paths containing large number of edges of very small length.

With $X' = X^*$ being an optimal solution of cost $\mathrm{OPT}$ and $\alpha = 1$, the bound in (1) is equal to $2\mathrm{OPT}$, which corresponds to the conventional algorithm which ignores the predictions. With $\alpha$ approaching infinity, its limit is $\mathrm{OPT} + \eta^+ + \eta^-$, matching the result from Theorem 2. However, it can be much better than both in case of $\widehat{X} \setminus X^*$ containing a small number of edges of very high weight. Compared to Theorem 2, $\eta^+$ in (1) is capped by $(\alpha - 1) \sum_{e \in S} \mathbf{c}(e)$ where $\sum_{e \in S} \mathbf{c}(e) \leqslant \mathbf{c}(\mathrm{MST}) \leqslant 2w(X^*)$, since Mehlhorn's algorithm is a 2-approximation algorithm for Minimum Steiner Tree. Since we always have $\eta^- \leqslant w(X^*)$, (1) shows that Algorithm 2 is a $2\alpha$-approximation algorithm regardless of the prediction error.

The key part of the proof of Theorem 7 is the analysis of how edges in $\widehat{X} \cap X^*$ and edges in $\widehat{X} \setminus X^*$ respectively influence $\mathrm{MST}_\alpha$ found by Algorithm 2, depending on the parameter $\alpha$. The improvement in the approximation ratio then comes from the short-cutting procedure on the graph with scaled weights which identifies paths over edges which are useful for connecting a higher number of terminals.

Consider a fixed Steiner tree $X' \subseteq E$. For the purpose of analysis, we define another weight

function $w'_\alpha$. We set

$$w'_\alpha(e) := \begin{cases} w(e)/\alpha & \text{if } e \in \widehat{X} \cap X', \\ w(e) & \text{otherwise.} \end{cases}$$

Denote by $\text{MST}'_\alpha$ a minimum spanning tree in $\mathbf{G}$ with respect to $\mathbf{c}'_\alpha$, i.e., the metric closure of $w'_\alpha$. The following observation holds.

**Observation 8.** *The cost of the connections in $\text{MST}'_\alpha$ can be bounded as*

$$\mathbf{c}'_\alpha(\text{MST}'_\alpha) \leqslant 2w(X') - 2(1 - 1/\alpha)w(\widehat{X} \cap X'). \tag{2}$$

*Proof.* In the graph $(V, X')$, replace each edge $uv \in X'$ by two directed edges $(u, v)$ and $(v, u)$, each of weight $w(uv)$. This way, each vertex has the same in-degree as out-degree and hence, by Euler's theorem (Korte and Vygen, 2012, Theorem 2.24), there is a tour $P$ using each directed edge exactly once. Since $X'$ is a Steiner tree, this tour visits all the terminals and naturally defines a spanning tree on $\mathbf{G}$ of cost at most $w(P)$: Whenever $P$ visits a new terminal $t$, we add an edge $tt' \in \mathbf{G}$ of cost $\mathbf{c}(tt') = w(p(tt'))$, where $t'$ is the preceding terminal visited by $P$ and $p(tt')$ is the segment of $P$ connecting $t'$ and $t$.

The weight of the tour $P$ is $2w(X')$ with respect to $w$ and $2w(X') - (1 - 1/\alpha)w(\widehat{X} \cap X')$ with respect to $w'_\alpha$, since every edge $e \in \widehat{X} \cap X'$ is used twice and has cost $w'_\alpha(e) = w(e) - (w(e) - w(e)/\alpha)$. Therefore, we have $\mathbf{c}(\text{MST}'_\alpha) \leqslant 2w(X') - (1 - 1/\alpha)w(\widehat{X} \cap X')$. $\qquad\square$

The following basic fact about spanning trees follows from the exchange property of matroids which states that for any two spanning trees $T_1$ and $T_2$ and any $e \in T_1 \setminus T_2$, there is $e' \in T_2 \setminus T_1$ such that $(T_1 \setminus \{e\}) \cup \{e'\}$ is a spanning tree. For a proof of the exchange property, see for instance Korte and Vygen (2012, Theorem 14.7).

**Proposition 9.** *Consider a minimum spanning tree $T$ on a graph $G$ with cost function $c\colon E \to \mathbb{R}_{\geqslant 0}$ and an arbitrary spanning tree $T'$ on the same graph. There exists a bijection $\phi\colon T \to T'$ such that $c(e) \leqslant c(\phi(e))$ for each edge $e \in T$.*

Proof of the following lemma contains the key part of our analysis. It uses Proposition 9 to charge each edge in $\widehat{X} \setminus X'$ to a single connection in $\text{MST}'_\alpha$.

**Lemma 2.** *Let $S'$ be a set of $\min\{k - 1, |\widehat{X} \setminus X'|\}$ edges with highest cost $\mathbf{c}'_\alpha$ in $\text{MST}'_\alpha$. We have*

$$w'_\alpha(X) \leqslant \mathbf{c}'_\alpha(\text{MST}'_\alpha) + \min\left\{\eta^+, (\alpha - 1)\sum_{e' \in S'} \mathbf{c}'_\alpha(e')\right\}.$$

*Proof.* Denote by $e'_1, e'_2, \ldots, e'_{k-1}$ the edges of $\text{MST}'_\alpha$ and by $e_1, e_2, \ldots, e_{k-1}$ the edges of $\text{MST}_\alpha$. Since $\text{MST}_\alpha$ is a minimum spanning tree with respect to $c_\alpha(\cdot)$ and up to reordering the edges, by Proposition 9, we can assume that

$$c_\alpha(e_i) \leqslant c_\alpha(e'_i), \quad \forall i \in \{1, \ldots, k - 1\}.$$

Let $F_{\leqslant 0} := \emptyset$ and set $F_{\leqslant i} := F_{\leqslant i-1} \cup (p(e_i) \setminus F_{\leqslant i-1})$, for $i = \{1, \ldots, k - 1\}$, so that $F = F_{k-1}$.

To show the lemma, for each pair $e_i, e'_i$, we distinguish between two cases:

1. $(\boldsymbol{p(e_i) \setminus F_{\leqslant i-1}}) \cap (\boldsymbol{\hat{F} \setminus F^*}) = \boldsymbol{\emptyset}$. In this case, no weights of edges on $(p(e_i) \setminus F_{i-1}) \cap (\hat{F} \setminus F^*)$ have been scaled down, hence

$$c'_\alpha(p(e_i) \setminus F_{\leqslant i-1}) = c_\alpha(p(e_i) \setminus F_{i-1}) \leqslant c_\alpha(e'_i) \leqslant c'_\alpha(e'_i).$$

2. $(p(e_i) \setminus F_{\leqslant i-1}) \cap (\hat{F} \setminus F^*) \neq \emptyset$. In this case, the actual cost of the path corresponding to edge $e_i$ minus the edges in $F_{\leqslant i-1}$ already bought, i.e. $p(e_i) \setminus F_{\leqslant i-1}$, can be at most $\alpha$ times higher:

$$c'_\alpha(p(e_i) \setminus F_{\leqslant i-1}) \leqslant \alpha \cdot c_\alpha(e_i) \leqslant \alpha \cdot c_\alpha(e'_i) \leqslant c'_\alpha(e'_i) + (\alpha - 1) \cdot c'_\alpha(e'_i).$$

At the same time, we have

$$c'_\alpha(p(e_i) \setminus F_{\leqslant i-1}) \leq c_\alpha(e_i) + c(\hat{F} \cap (p(e_i) \setminus F_{\leqslant i-1})) \leqslant c'_\alpha(e'_i) + c(\hat{F} \cap (p(e_i) \setminus F_{\leqslant i-1})).$$

Since the latter case only happens once for each edge in $\hat{F} \setminus F^*$, we see that

$$c'_\alpha(F) \leqslant \underbrace{\sum_{i=1}^{k-1} c'_\alpha(e'_i)}_{c'_\alpha(\mathrm{MST}'_\alpha)} + (\alpha - 1) \max_{S \subseteq [k]: |S| = |\hat{F} \setminus F^*|} \sum_{i \in S} c'_\alpha(e'_i),$$

and at the same time, we have $c'_\alpha(F) \leqslant \mathrm{MST}'_\alpha + c(\hat{F})$. The lemma follows. $\qquad \square$

*Proof of Theorem 7.* Since for all edges $e \in \widehat{X} \cap X'$ we have $w'_\alpha(e) = w(e)/\alpha$, we can write $w(e) = w'_\alpha(e) + (1 - 1/\alpha)w(e)$. Therefore, we have

$$w(X) \leqslant w'_\alpha(X) + (1 - 1/\alpha) \cdot w(\widehat{X} \cap X').$$

Combining this bound with Lemma 2 and (2), we get

$$w(X) \leqslant 2w(X') - (1 - 1/\alpha)w(\widehat{X} \cap X') + \min\left\{\eta^+, (\alpha - 1) \sum_{e' \in S'} \mathbf{c}'_\alpha(e')\right\}.$$

Proposition 9 implies that $\sum_{e' \in S'} c'_\alpha(e') \leq \sum_{e \in S} c(e)$, since $\mathrm{MST}'_\alpha$ is a minimum spanning tree on $\mathbf{G}$ with respect to cost $c'_\alpha$ and $\mathrm{MST}$ is a spanning tree on the same graph. Now, it is enough to note that

$$2w(X') - (1 - 1/\alpha)w(\widehat{X} \cap X') = (1 + 1/\alpha)w(X') + (1 - 1/\alpha)w(X' \setminus \widehat{X}). \qquad \square$$

## 4.2 Approximating the best alpha

Let $\alpha^*$ be the parameter of Algorithm 2 which minimizes the upper bound in Theorem 7. We show how to find $\alpha$ which achieves a bound at most $(1 + \epsilon)$ times worse than the bound with $\alpha^*$.

---

**Algorithm 3:** Steiner tree: search for the best $\alpha$

---

**for** $i = 0, \ldots, \lceil \log_{1+\epsilon} \epsilon^{-1} \rceil$ **do**
$\quad$ run Algorithm 2 with $\alpha = (1 + \epsilon)^i$;
output the best solution found during above iterations;

---

**Theorem 10.** *For a constant $\epsilon > 0$, Algorithm 3 runs in near-linear time and finds a Steiner tree $X$ with weight*

$$w(X) \leqslant (1 + \epsilon) \min_{\alpha \geqslant 1}\left\{\left(1 + \frac{1}{\alpha}\right) w(X') + \left(1 - \frac{1}{\alpha}\right) \eta^- + \min\left\{\eta^+, (\alpha - 1) \cdot \sum_{e \in S} \mathbf{c}(e)\right\}\right\}.$$

*Proof.* Algorithm 3 performs a constant number $\lceil \log_{1+\epsilon} \epsilon^{-1} \rceil + 1$ of iterations of Algorithm 2, which runs in near-linear time.

First, note that we need to consider only $\alpha \leqslant \epsilon^{-1}$, because

$$(1+\epsilon)c(X') + (1-\epsilon)\eta^- + \min\left\{\eta^+, (\epsilon^{-1}-1)\cdot\sum_{e\in S}\mathbf{c}(e)\right\}\right) \leqslant (1+\epsilon)f(\alpha)$$

for any $\alpha > \epsilon^{-1}$.

It is enough to show that, for any $\alpha$, we have $f((1+\epsilon)\alpha) \in [(1+5\epsilon)^{-1}f(\alpha), (1+5\epsilon)f(\alpha)]$. We show this for every term of $f(\alpha)$ separately. We have

$$\left(1+\frac{1}{\alpha}\right)c(X') \geqslant \left(1+\frac{1}{(1+\epsilon)\alpha}\right)c(X') = \frac{(1+\epsilon)\alpha+1}{(1+\epsilon)\alpha}c(X') \geqslant (1+\epsilon)^{-1}\left(1+\frac{1}{\alpha}\right)c(X').$$

Similarly, we have

$$\left(1-\frac{1}{\alpha}\right)\eta^- \leqslant \left(1-\frac{1}{(1+\epsilon)\alpha}\right)\eta^- \leqslant \left(\frac{(1+\epsilon)\alpha-1}{(1+\epsilon)\alpha}\right)\eta^- \leqslant (1+\epsilon)\left(\frac{\alpha-1}{(1+\epsilon)\alpha}\right)\eta^-,$$

which is at most $(1+\epsilon)(1-1/\alpha)\eta^-$. For the last term, we have

$$\alpha - 1 \leqslant (1+\epsilon)\alpha - 1 = \alpha - 1 + \epsilon\alpha \leqslant (1+2\epsilon)(\alpha-1)$$

if $\alpha \geqslant 2$. If $\alpha < 2$, we have $\epsilon\alpha\sum_{e\in S}\mathbf{c}(e_i) \leqslant 4\epsilon c(X')$. $\qquad\square$

## 4.3   Tight example for our analysis

Figure 1 describes an instance with $k = n - 1$ terminals and a prediction with a single false-negative edge of weight $1 + \epsilon$ and a single false-positive edge of weight $\beta > 2$. Algorithm 2 with $\alpha \in [1, (1+\epsilon)^{-1})$ achieves approximation ratio approaching 2 as $n$ increases. With $\alpha \in (\frac{1}{1+\epsilon}, \frac{\beta}{1+\epsilon})$, its approximation ratio is equal to 1. With $\alpha > \frac{\beta}{1+\epsilon}$, it approaches $\frac{\mathrm{OPT}+\beta}{\mathrm{OPT}}$, which is equal to 2 if we choose $\beta = \mathrm{OPT}$. This shows that the best choice of $\alpha$ may not be 1 nor approaching $\infty$.

We can use this construction to show that the coefficient of $\sum_{e\in S}\mathbf{c}(e)$ in (1) is tight for the given algorithm. First consider the input graph in Figure 1 with $\beta = 2n$. Algorithm 2 with $\alpha = \beta$ receiving a prediction $\widehat{X}$ containing all red edges, achieves cost $(n-1)(1+\epsilon)+\beta \geqslant \mathrm{OPT}+\frac{\alpha}{2}2-(1+\epsilon)$, where 2 is the cost of the most expensive connection in the minimum spanning tree on the metric closure of the terminals. Note that all terms except for $-(1+\epsilon)$ go to infinity as $n$ increases.

Similarly, we can show the tightness of the coefficient of $\eta^-$. Algorithm 2 with $\widehat{X} = \emptyset$ achieves cost $2n \geqslant (1+\epsilon)^{-1}\big((1+1/\alpha)\mathrm{OPT}+\eta^-$, since $\eta^- = \mathrm{OPT} = n(1+\epsilon)$.

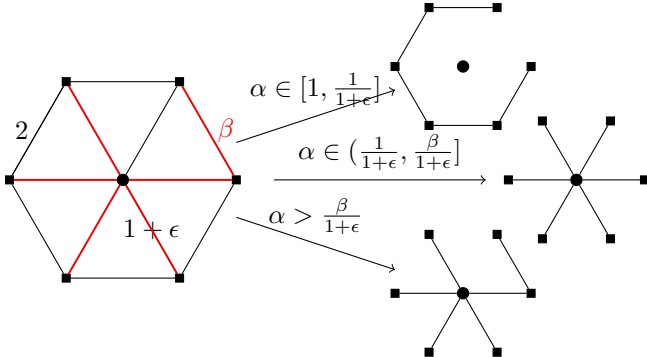

Figure 1: Steiner tree problem with $k = n - 1$ terminals (square) and one non-terminal vertex (circle). Outer edges have weight 2, except one edge of weight $\beta$. Inner edges have weight $1 + \epsilon$. Red edges are predicted. On the right are the respective Steiner trees output by Algorithm 2 with different parameters of $\alpha$.

# 5 Lower bounds

## 5.1 Preliminaries

In order to obtain lower bounds for our setting we use the following results of Khot and Regev (2008).

**Proposition 11** (Khot and Regev (2008)). *Assuming UGC, for every constant $\delta > 0$, it is NP-hard to distinguish, for an input $n$-vertex graph $G$, between the following two cases:*

- *(YES) $G$ contains an independent set of size at least $\frac{n}{2} - \delta n$.*

- *(NO) $G$ contains no independent set of size at least $\delta n$.*

It directly implies the following corollary for the Minimum Vertex Cover problem:

**Corollary 12** (Khot and Regev (2008)). *Assuming UGC, for every constant $\delta > 0$ it is NP-hard to distinguish, for an input $n$-vertex graph $G$, between the following two cases:*

- *(YES) $G$ has a vertex cover of size at most $\frac{n}{2} + \delta n$.*

- *(NO) $G$ has no vertex cover of size at most $(1 - \delta)n$.*

Note that Corollary 12 implies that, assuming UGC and P$\neq$NP, Minimum Vertex Cover cannot be approximated within a factor of $2 - \epsilon$, for any $\epsilon > 0$.

## 5.2 Lower bound for a minimization problem

We now argue that our Theorem 2 for minimization selection problems cannot be improved. For a specific minimization selection problem, namely the Minimum Vertex Cover problem, which has a folklore 2-approximation algorithm, Theorem 2 and Corollary 3 with $\rho = 2$ imply existence of a learning-augmented algorithm with approximation ratio $\min\{1 + \frac{\eta^+ + \eta^-}{\text{OPT}}, 2\}$. We show that this is the best possible upper bound.

**Theorem 13.** *Let $A$ be a polynomial-time learning-augmented approximation algorithm for Minimum Vertex Cover that guarantees an approximation ratio of at most*

$$1 + f\left(\frac{\eta^+}{\text{OPT}}, \frac{\eta^-}{\text{OPT}}\right),$$

*for some non-decreasing function $f$. Then, assuming UGC, we have*

$$f(x, y) \geqslant x + y,$$

*for any choice of $x, y \geqslant 0$ satisfying $x + y \leqslant 1$.*

*Proof.* By contradiction, consider $x, y$ such that $f(x, y) < x + y$. We denote $\epsilon := (x + y) - f(x, y)$, and let $\delta := \epsilon/20$. We will show how to use the hypothesized learning-augmented algorithm $A$ to construct an algorithm $\bar{A}$ for the (prediction-less) Minimum Vertex Cover problem that can distinguish between instances on $n$ vertices with optimum value more than $(1 - \delta)n$ and instances with optimum value at most $(1 + \delta)\frac{n}{2}$. Such an algorithm would contradict Corollary 12. In order to provide a cleaner exposition, we first assume that $A$ can actually solve *Min-Weight* Vertex Cover with the given approximation ratio. This assumption can be removed easily, as discussed at the end of the proof.

We construct algorithm $\bar{A}$ that, for any input graph $G$ with minimum vertex cover $X^*$ of size $|X^*| \in \left[(1 - \delta)\frac{n}{2}, (1 + \delta)\frac{n}{2}\right]$, produces a solution $X$ such that $|X| < (1 - \delta)n$, certifying that $G$ does not belong to the NO case. Note that for graphs with vertex cover of size $< (1 - \delta)\frac{n}{2}$ such solution can be easily produced by the folklore 2-approximation algorithm.

Given the input graph $G$, $\bar{A}$ produces a weighted graph $\bar{G}$ and a prediction $\hat{X}$, and runs $A$ on $\bar{G}$ with $\hat{X}$. Graph $\bar{G}$ is a disjoint union of three graphs: $G_0$, $G_+$, and $G_-$. Let $x' := \frac{1-\delta}{1+\delta}x$ and $y' := \frac{1-\delta}{1+\delta}y$.

- $G_0$ is a complete graph on $\frac{n}{2} + 1$ vertices, we have $\bar{w}(v) = (1 - x' - y')$ for each $v \in V(G_0)$.

- $G_+$ is a copy of $G$ with scaled weights, we have $\bar{w}(v) = x'$ for each $v \in V(G_+)$.

- $G_-$ is a copy of $G$ with scaled weights, we have $\bar{w}(v) = y'$ for each $v \in V(G_-)$.

The prediction $\hat{X}$ contains arbitrary $\frac{n}{2}$ vertices from $G_0$, all vertices of $V(G_+)$ and no vertices of $V(G_-)$. Receiving a solution $\bar{X}$ found by the algorithm $A$ on $\bar{G}$, we denote $X_+ := \bar{X} \cap V(G_+)$ and $X_- := \bar{X} \cap V(G_-)$. Algorithm $\bar{A}$ returns $X$ which is either $X_+$ or $X_-$, choosing the one with the smaller size. We now prove that $|X| < (1 - \delta)n$ if $G$ is the YES case.

First, we show that $\eta^+/\mathrm{OPT} \leqslant x$ and $\eta^-/\mathrm{OPT} \leqslant y$. Let OPT be the value of an optimal solution for $\bar{G}$. We have $\mathrm{OPT} = (1 - x - y)\frac{n}{2} + x|X^*| + y|X^*|$ which belongs to $\left[(1-\delta)\frac{n}{2}, (1+\delta)\frac{n}{2}\right]$ by the assumption about the size of $X^*$. For the error, we have

$$\eta^+ = x' \cdot (n - |X^*|) = \frac{1-\delta}{1+\delta} x \cdot (n - |X^*|), \quad \text{and}$$

$$\eta^- = y' \cdot |X^*| = \frac{1-\delta}{1+\delta} y \cdot |X^*|.$$

Since both $(n - |X^*|)$ and $|X^*|$ are at most $(1+\delta)\frac{n}{2}$ and $\mathrm{OPT} \geqslant (1-\delta)\frac{n}{2}$, we have $\eta^+/\mathrm{OPT} \leqslant x$ and $\eta^-/\mathrm{OPT} \leqslant y$. By monotonicity of $f$, this implies that $\bar{X}$ is a $(1 + x + y - \epsilon)$-approximate solution. Therefore, we have

$$
\begin{aligned}
\bar{w}(\bar{X}) &\leqslant (1 + x + y - \epsilon) \cdot \mathrm{OPT} \\
&\leqslant (1 + x + y) \cdot (1+\delta)\frac{n}{2} - \epsilon \cdot (1-\delta)\frac{n}{2} \\
&\leqslant \frac{n}{2} + (x+y)\frac{n}{2} + (1 + x + y + \epsilon)\delta\frac{n}{2} - \epsilon\frac{n}{2} \\
&\leqslant \frac{n}{2} + (x+y)\frac{n}{2} + 4\delta\frac{n}{2} - \epsilon\frac{n}{2} \\
&\leqslant \frac{n}{2} + (x+y)\frac{n}{2} - 0.8\epsilon\frac{n}{2},
\end{aligned}
\tag{3}
$$

by our choice of $\delta = \epsilon/20$. On the other hand, we can write

$$
\begin{aligned}
\bar{w}(\bar{X}) &\geqslant (1 - x' - y')\frac{n}{2} + x'|X_+| + y'|X_-| \\
&= \frac{n}{2} + x'\left(|X_+| - \frac{n}{2}\right) + y'\left(|X_-| - \frac{n}{2}\right) \\
&\geqslant \frac{n}{2} + x\left(|X_+| - \frac{n}{2}\right) + y\left(|X_-| - \frac{n}{2}\right) - 4\delta\frac{n}{2} \\
&\geqslant \frac{n}{2} + x\left(|X_+| - \frac{n}{2}\right) + y\left(|X_-| - \frac{n}{2}\right) - 0.2\epsilon\frac{n}{2}.
\end{aligned}
\tag{4}
$$

In the second to last step, we use the definition of $x'$ and $y'$, and the fact that $\frac{1-\delta}{1+\delta} \geqslant 1 - 2\delta$. Inequalities (3) and (4) imply that at least one of $|X_+| - \frac{n}{2}$ and $|X_-| - \frac{n}{2}$ must be less than or equal to $\frac{n}{2} - 0.6\epsilon\frac{n}{2} < \frac{n}{2} - \delta n$. Indeed, otherwise we would have

$$\bar{w}(\bar{X}) > \frac{n}{2} + (x+y)\frac{n}{2} - (x+y)0.6\epsilon\frac{n}{2} - 0.2\epsilon\frac{n}{2} \geqslant \frac{n}{2} + (x+y)\frac{n}{2} - 0.8\epsilon\frac{n}{2},$$

contradicting (3). In other words, we have either $|X_+| < (1 - \delta)n$ or $|X_-| < (1 - \delta)n$, which concludes the proof.

In order to transform $\bar{G}$ into an unweighted instance, we take (roughly) $1/x'y'$ copies of $G_0$, $1/(1 - x' - y')y'$ copies of $G_+$ and $1/(1 - x' - y')x'$ copies of $G_-$, each vertex having weight 1. Prediction contains $n/2$ vertices from each copy of $G_0$, all vertices of all copies of $G_+$ and no vertices from the copies of $G_-$. This is to ensure that the minimum vertex cover on the copies of $G_0$, on the copies of $G_+$, and on the copies of $G_-$ respectively have approximately the same ratio as in $\bar{G}$. In order to achieve approximation ratio $1 + x + y - \epsilon$, the algorithm $A$ would need to find a solution to at least one copy of $G_-$ or $G_+$ with size smaller than $(1 - \delta)n$. $\qquad\square$

## 5.3 Lower bound for a maximization problem

We show a similar result for the class of considered maximization problems.

**Theorem 14.** *Let $A$ be a polynomial-time learning-augmented approximation algorithm for Maximum Independent Set that guarantees an approximation ratio of at least*

$$1 - f\left(\frac{\eta^+}{\text{OPT}}, \frac{\eta^-}{\text{OPT}}\right),$$

*for some non-decreasing function $f$. Then, assuming UGC, we have*

$$f(x, y) \geqslant x + y,$$

*for any choice of $x, y \geqslant 0$ satisfying $x + y \leqslant 1$.*

*Proof.* By contradiction, consider $x, y$ such that $f(x, y) < x + y$. We denote $\epsilon := (x + y) - f(x, y)$, and let $\delta := \epsilon/20$. We proceed similarly to the proof of Theorem 13, and we reach contradiction with Proposition 11.

We construct algorithm $\bar{A}$ such that for any input graph $G$ with maximum independent set $X^*$ of size $|X^*| \in [(1 - \delta)\frac{n}{2}, (1 + \delta)\frac{n}{2}]$ it produces an independent set $X$ of size $|X| > \delta n$, certifying that $G$ does not belong to the NO case. Note that graphs with an independent set larger than $(1 + \delta)\frac{n}{2}$ have a minimum vertex cover of size smaller than $(1 - \delta)\frac{n}{2}$. Therefore, in such graphs we can find a 2-approximate solution $C$ of the minimum vertex cover problem and then $V(G) \setminus C$ will be an independent set of size $n - |C| > n - (1 - \delta)n = \delta n$.

Given the input graph $G$, $\bar{A}$ constructs a (weighted) graph $\bar{G}$ and a prediction $\hat{X}$, and uses them as an input for algorithm $A$. Graph $\bar{G}$ is a disjoint union of three graphs: $G_0$, $G_+$, and $G_-$. Let $x' := \frac{1-\delta}{1+\delta}x$ and $y' := \frac{1-\delta}{1+\delta}y$.

- $G_0$ is a graph with $n/2$ vertices and no edges, we have $\bar{w}(v) = (1 - x' - y')$ for each $v \in V(G_0)$.

- $G_+$ is a copy of $G$ with scaled weights, we have $\bar{w}(v) = x'$ for each $v \in V(G_+)$.

- $G_-$ is a copy of $G$ with scaled weights, we have $\bar{w}(v) = y'$ for each $v \in V(G_-)$.

The prediction $\hat{X}$ contains all vertices from $G_0$, all vertices of $V(G_+)$ and no vertices of $V(G_-)$. Receiving a solution $\bar{X}$ found by algorithm $A$ on $\bar{G}$, we denote $X_+ = \bar{X} \cap V(G_+)$ and $X_- = \bar{X} \cap V(G_-)$. $\bar{A}$ returns the larger of the two sets $X_+$ and $X_-$, which we denote by $X$. We now prove that $|X| > \gamma n$ if $G$ is the YES case.

First, we show that $\eta^+/\text{OPT} \leqslant x$ and $\eta^-/\text{OPT} \leqslant y$. Let OPT be the value of an optimal solution on $\bar{G}$. We have $\text{OPT} = (1 - x - y)\frac{n}{2} + x|X^*| + y|X^*|$ which belongs to $[(1 - \delta)\frac{n}{2}, (1 + \delta)\frac{n}{2}]$ by the assumption about the size of $X^*$. For the error, we have

$$\eta^+ = x' \cdot (n - |X^*|) = \frac{1 - \delta}{1 + \delta}x \cdot (n - |X^*|), \quad \text{and}$$

$$\eta^- = y' \cdot |X^*| = \frac{1 - \delta}{1 + \delta}y \cdot |X^*|.$$

Since both $(n - |X^*|)$ and $|X^*|$ are at most $(1 + \delta)\frac{n}{2}$ and $\text{OPT} \geqslant (1 - \delta)\frac{n}{2}$, we have $\eta^+/\text{OPT} \leqslant x$ and $\eta^-/\text{OPT} \leqslant y$. By monotonicity of $f$, this implies that $\bar{X}$ is at least a $(1 - x - y + \epsilon)$-approximate solution. Therefore, we have

$$\bar{w}(\bar{X}) \geqslant (1 - x - y + \epsilon) \cdot \text{OPT}$$
$$\geqslant (1 - x - y) \cdot (1 - \delta)\frac{n}{2} + \epsilon \cdot (1 - \delta)\frac{n}{2}$$
$$\geqslant (1 - x - y)\frac{n}{2} - 3\delta\frac{n}{2} + \epsilon\frac{n}{2}.$$

On the other hand, we can write

$$\bar{w}(\bar{X}) \leqslant (1 - x' - y')\frac{n}{2} + x'|X_+| + y'|X_-|$$
$$\leqslant (1 - x - y)\frac{n}{2} + x'|X_+| + y'|X_-| + 4\delta\frac{n}{2}.$$

In the last step, we use the definition of $x'$ and $y'$, and $\frac{1-\delta}{1+\delta} \geqslant 1 - 2\delta$. The two equations above imply that at least one of $|X_+|$ and $|X_-|$ must be larger than $(\epsilon\frac{n}{2} - 7\delta\frac{n}{2})/2 > \delta n$, by our choice of $\delta$. In other words, we have that $|X_+| > \delta n$ or $|X_-| > \delta n$, which concludes the proof.

In order to transform $\bar{G}$ into an unweighted instance, we take $\frac{1}{x'y'}$ copies of $G_0$, $\frac{1}{(1-x'-y')y'}$ copies of $G_+$ and $\frac{1}{(1-x'-y')x'}$ copies of $G_-$, each vertex having weight 1. □

## 6 Experimental evaluation

In this section we present an experimental evaluation of our refined learning-augmented algorithm for the Steiner Tree problem from Section 4. The source code is available on GitHub[4].

**Dataset.**  We base our experiments on the heuristic track of the 2018 PACE Challenge (Bonnet and Sikora, 2018), which is to our knowledge the most recent large-scale computational challenge concerning the Steiner Tree problem. In particular, we use their dataset[5], which consists of 199 graphs selected from among the hardest instances in the SteinLib library (Koch et al., 2000).[6]

**Algorithms.**  We also use PACE as the source of the state-of-the-art Steiner Tree solver, with which we compare our algorithm. More specifically, in our experiments we evaluate three algorithms:

- **Mehlhorn**'s 2-approximation algorithm, implemented by us in C++. Its mean empirical approximation factor on PACE instances that we observe is $\approx 1.17$. On average it spends 34 milliseconds per instance, and never needs more than 500 milliseconds.

- **CIMAT**, the winning solver from the PACE Challenge, whose C++ implementation is available on GitHub (Ruiz et al., 2018). It is an evolutionary algorithm that can be stopped at any time and outputs the best solution found so far. It was designed to run for 30 minutes per instance, as this was the time limit in PACE. However, we noticed that on 95% of instances already after one minute it produces a solution within 1.01 of the optimum, and therefore we decided to always run it only for one minute, in order to keep the computational costs of the experiments down. We also remark that, on average, it took the CIMAT solver $\approx 10$ seconds to output a first feasible solution. In all our experiments, we use the value of the solution returned by CIMAT as an estimate of the value of an optimal solution OPT, which would be difficult and impractical to calculate exactly.

- **ALPS**, our **al**gorithm with **p**redictions from Section 4, with the "confidence" hyperparameter $\alpha \in \{1.1, 1.2, 1.4, 2, 4, \infty\}$. In terms of efficiency it is virtually indistinguishable from Mehlhorn's algorithm, which it runs as a subroutine and which dominates its running time.

**Predictions.**  We generate both synthetic predictions and learned predictions. For both types of predictions we vary their quality, which is captured by the parameter $p \in \{0.0, 0.1, \ldots, 1.0\}$; the higher the parameter $p$ the higher the prediction error. The predictions are then fed to our algorithm (ALPS), and we compare the quality of its outputs with those of the other two algorithms (CIMAT and Mehlhorn's).

- **Synthetic predictions.** In our first experiment, we simulate a predictor by introducing artificial noise to groundtruth labels. For each instance from the dataset, we compute a (near-)optimal

---

[4]Available at `github.com/adampolak/steiner-tree-with-predictions`.

[5]Available at `github.com/PACE-challenge/SteinerTree-PACE-2018-instances`.

[6]As of 2018, a majority of these instances could not be solved to optimality within one hour with state-of-the-art solvers. The average number of vertices is $\approx 27$k, the average number of edges is $\approx 48$k and the average number of terminals is $\approx 1$k.

solution $S \subseteq E$ using CIMAT. Then, for each value of the parameter $p$, we swap out a randomly selected $p$-fraction of the edges in $S$ for a randomly selected subset of edges from $E \setminus S$ of the same cardinality. This yields a prediction with error roughly $\eta^+ \approx p \cdot \mathrm{OPT}$ and $\eta^- \approx p \cdot \mathrm{OPT}$.

- **Learned predictions.** In our second, more realistic experiment we test our approach end-to-end, i.e., we first learn predictions, and then use them to solve instances not seen during learning. For this experiment, we need to work with *distributions over instances* instead of individual problem instances. To this end, for each instance $I = (V, E, T, w)$ from our dataset, and for each value of the parameter $p$, we consider a distribution that returns instances of the form $I' = (V, E, T', w)$, with a $p$-fraction of the terminals resampled (but with the same underlying graph and edge weights). Actually, we consider two types of distributions, depending on how they resample terminals:

    - **"Fixed core" distribution.** We fix a $(1-p)$-fraction of the terminals $T_{\mathrm{old}} \subseteq T$ and for each sampled instance $I'$ we swap out the remaining $p$-fraction by independently sampling $\lceil p \cdot |T| \rceil$ terminals $T_{\mathrm{new}} \subseteq V \setminus T$, returning $I' = (V, E, T_{\mathrm{old}} \cup T_{\mathrm{new}}, w)$.
    - **"No core" distribution.** Each time we sample a fresh $(1-p)$-fraction of the terminals in $T$ to obtain $T_{\mathrm{old}}$ (i.e., $T_{\mathrm{old}} \subseteq T$ is not fixed over all the samples).

For each such distribution, we sample $k = 10$ instances, and we compute a (near-)optimal solution to each of them using CIMAT. Then, we evaluate ALPS on each such sampled instance using the prediction learned from (the solutions to) the remaining $k - 1 = 9$ instances sampled from the same distribution (i.e., we preform leave-one-out cross-validation). To learn the prediction from a set of left-out instances, we predict edge $e \in E$ iff it appeared in more than half of the (near-)optimal solutions to the left-out instances, which coincides with empirical risk minimization of the combined prediction error $\eta_+ + \eta_-$.

**Evaluation metrics.** To evaluate the performance of our algorithms, we use two different measures. For the synthetic predictions we look at the (empirical) approximation ratios observed for each of the algorithms and averaged over all the instances (see Figure 2a). For the learned predictions such average would not be a useful metric – this is because swapping out a set of terminals often completely changes both the value of an optimal solution and the relative performance of Mehlhorn's algorithm (see Figures 2b–2d). For this reason, before averaging over the instances, we normalize the solutions costs so that $0$ corresponds to the optimum and $1$ corresponds to the performance of Mehlhorn's algorithm. Specifically, denoting by $c_{\mathrm{ALPS}(\alpha)}, c_{\mathrm{OPT}}$, and $c_{\mathrm{MST}}$ the cost of the solution output by ALPS (with parameter $\alpha$), the optimum value and the cost output by Mehlhorn's algorithm, respectively, we define the normalized cost of ALPS as $\frac{c_{\mathrm{ALPS}(\alpha)} - c_{\mathrm{OPT}}}{c_{\mathrm{ALPS}(\alpha)} - c_{\mathrm{MST}}}$. For predictions of varying quality $p$ and for different values of $\alpha$, this measure then accurately reflects how ALPS interpolates between the optimum ($0$) and Mehlhorn's algorithm ($1$).

**Results.** The results of our empirical evaluation are depicted in Figure 2. Figure 2a displays the performance of ALPS with synthetic prediction, averaged over all instances. Figures 2b, 2c, and 2d depict several typical behaviors on distributions of the Steiner Tree instances with resampled terminals, which we used in our experiment with learned predictions. Those figures illustrate the need to consider normalized costs. Finally, Figures 2e and 2f display the normalized costs of algorithms averaged over all distributions, with fixed core and with no core, respectively.

Out of the datapoints that we report, $95\%$ of them had standard deviations below $0.1$ of their values, and the maximum standard deviation among them was $0.25$ of the corresponding value. It would be interesting to run the experiments with a higher number of iterations in order to bring the standard deviations down, but our computational resources do not permit that (the described experiments already required $\approx 700$ CPU hours).

**Conclusions.** In all our experiments, ALPS (especially with the confidence parameter $\alpha \geqslant 1.4$) equipped with good predictions (low values of the parameter $p$, left side of the plots) outputs solutions almost as good as the (near-)optimal solutions output by CIMAT, which is orders of magnitude slower.

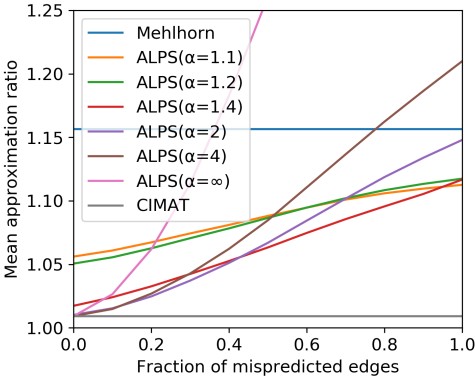

(a) Synthetic predictions. Average approximation ratio over all instances.

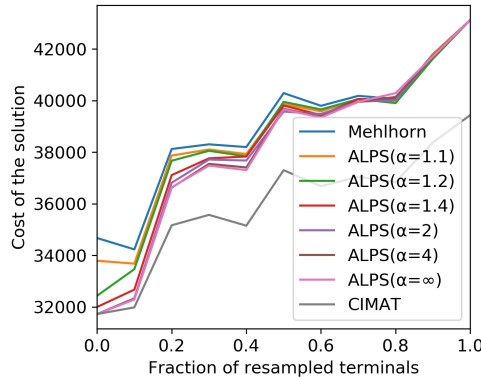

(b) Learned predictions. "Fixed core" distribution based on instance 011.

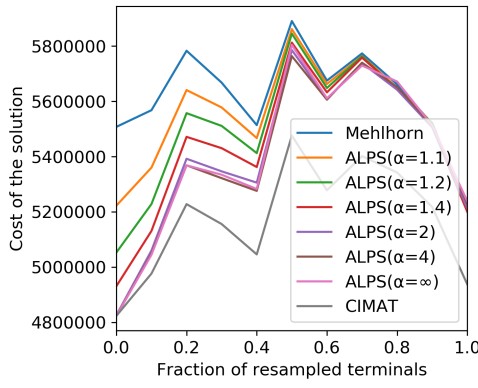

(c) Learned predictions. "No core" distribution based on instance 082.

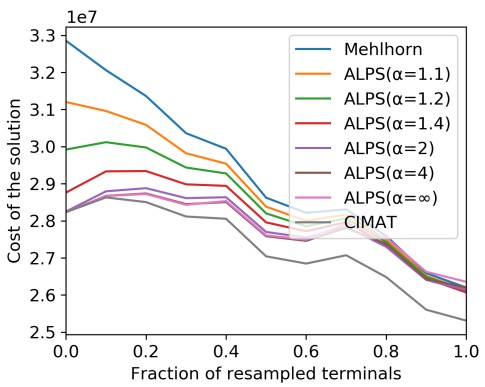

(d) Learned predictions. "Fixed core" distribution based on instance 178.

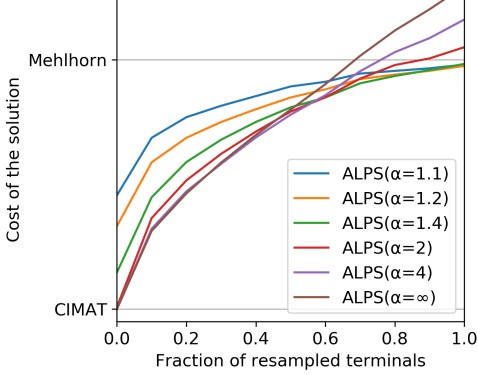

(e) Learned predictions. Normalized cost averaged over all distributions with fixed core.

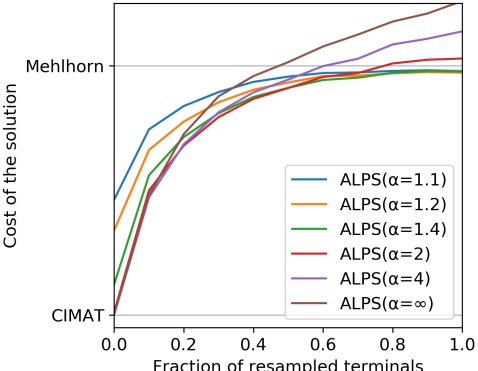

(f) Learned predictions. Normalized cost averaged over all distributions with no core.

Figure 2: Experimental evaluation of our refined Steiner Tree algorithm from Section 4 (ALPS), with different values of the confidence parameter $\alpha$, compared to an equally fast classic approximation algorithm (Mehlhorn) and a much slower near-optimal solver (CIMAT). The x-axis represents the parameter $p$ controlling the prediction error: accurate predictions are to the left and erroneous predictions are to the right. The y-axis represents the value of the returned solution (the lower the better).

On the other hand, the classic approximation algorithm of Mehlhorn, which has the same running time as ALPS, outputs solutions with a noticeably higher total cost.

Unsurprisingly, with the increasing prediction error (high values of the parameter $p$, right side of the plots) performance of ALPS slowly degrades. What is notable though is that already for moderate values of the confidence parameter ($\alpha \leqslant 2$) ALPS seems robust, i.e., it is never significantly worse than Mehlhorn, even with bad predictions. This is despite the fact that our ALPS implementation does not involve a separate robustification step (as in Corollary 3).

As the theory predicts, in the case of accurate predictions it is better to choose a large $\alpha$ while for bad predictions a smaller $\alpha$ is better, and, as we explain in Section 4.2, one may want to run ALPS for a geometric progression of $\alpha$'s and pick the best solution. However, in our experiments, it seems that choosing just a single $\alpha = 1.4$ or $\alpha = 2$ is a good choice almost universally.

Finally, it is worth to note that, somewhat surprisingly, synthetic predictions with $p = 1.0$, which are almost completely random, still allow (for small enough $\alpha$) to improve over Mehlhorn's algorithm. This surprising behavior can be explained by the fact that instances that are relatively hardest for the Mehlhorn's algorithm tend to be very symmetric and to have multiple (close-to-)optimal solutions, and it seems that randomly decreasing a small fraction of edge weights by a small factor $\alpha$ breaks the symmetry and guides the algorithm towards a good solution.

## 7 Discussion

We initiated the study of algorithms with predictions with a focus on improving over the approximation guarantees of classic algorithms without increasing the running time. This paper focused on the wide and important class of selection problems, but it would be interesting to investigate whether similar results can be obtained for central combinatorial optimization problems that do not belong to this class, e.g., clustering and scheduling problems or problems with non-linear (e.g., submodular) objectives.

We demonstrated, with the example of the Steiner Tree problem, that refined algorithms with improved guarantees are possible for specific problems. A second, implicit, advantage of our refined Steiner Tree algorithm is that its actual performance could be bounded in terms of a quantity directly related to the optimal cost, thus avoiding the additional robustification step. An interesting direction for further research would be to identify other problems satisfying these two properties.