# OpenReview forum: "Approximation algorithms for combinatorial optimization with predictions"
_ICLR.cc/2025/Conference — ICLR 2025 Spotlight_

### Official Review · Reviewer_9d1z · 2024-10-23

**Soundness:** 3
**Presentation:** 4
**Contribution:** 3
**Rating:** 8
**Confidence:** 3

**Summary:**

The paper is about learning-augmented approximation algorithms for combinatorial set selection problems. More specifically, the authors consider the (abstract) problem of selecting a set of elements of a universe of minimum total weight such that the selected set is feasible. A classic example of such a problem is the Vertex Cover problem. An approximation algorithm runs in polynomial time and computes a solution that is within an $\alpha$ factor of an optimal objective value. This factor is called the approximation factor of the algorithm. Typical applications for approximation algorithms are optimization problems that are NP-hard to solve optimally.
Moreover, for some problems such as Vertex Cover, finding a better-than-2 approximation algorithm would contradict standard complexity assumptions such as the Unique Games Conjecture or $P \neq NP$.

Learning-augmented algorithms are a recently popular method of beyond worst-case analysis, and are nowadays an established subfield in the intersection of algorithm theory and machine learning. The idea is to give an algorithm access to an additional input - a prediction - and analyze a learning-augmented algorithms performance w.r.t. the quality of this prediction. Only a few works have considered approximation algorithms under this framework so far.

The present paper considers the prediction model where a predicted solution is given to the algorithm. They present algorithms for the general set selection problem that achieve a near optimal performance for perfect predictions and a smooth degradation w.r.t. the number of false positive and false negatives of the prediction compared to some optimal solution.
At the same time, the approximation ratio of the currently best-known approximation algorithm can be achieved by running both algorithms and selecting the better solution.

Further results include:
- A similar result for maximization problems
- An algorithm with a controlable tradeoff between consistency and smoothness
- Lower bounds w.r.t. the Unique Games Conjecture showing that their algorithms are essentially best-possible.
- Empirical experiments

**Strengths:**

- The paper provides learning-augmented algorithms for various problems as well as general tight lower bounds on these results. The approximation guarantees in the case of good predictions improve over complexity lower bounds in the setting without predictions.
- There are only few papers on approximation algorithms with predictions. Thus, I have the feeling that its simplicity and strong results have an impact, and these results may be important for future works.
- The proposed algorithms are simple yet essentially best-possible.
- The paper is very well written, gives a structured overview of related work, and provides some empirical insights.

**Weaknesses:**

- One minor weakness is that the results are achieved using quite simple standard techniques. However, given that the analysis is essentially tight, and this is an AI conference and not a TCS conference, I think this is really only a minor issue.
- Compared to online algorithms with predictions, one can guarantee robustness for approximation algorithms for free by running both algorithms in parallel. Thus, there are no interesting insights between consistency and robustness for approximation algorithms. However, the authors show that similar trade-off are present between consistency and smoothness, give paramterized results and moreover discuss how to choose such parameters. Thus, I think that this is also only a minor weakness.
- I would have liked a concluding discussion about the impact of these results and a potential outlook at future questions at the end of the paper. This could be addressed in a camera-ready version though.
- In the "results" paragraph of the empirical evaluation, I was missing your takes on how your algorithm compares to CIMAT, given that it is part of your experiments. In general, I would have liked to see a larger discussion here. This could be also easily addressed in a revised version.

**Questions:**

Questions:
- L128: Can we conclude from this example that the linear depedence $(\rho - 1) \eta^-$ is necessary? As far as I understand, this example only gives lower bounds on the endpoints of the error functions, so in principle one could have a non-linear dependence.



Further comments on the writeup:
- L81: I think it would improve the readability if you explain before that the idea is to predict a solution (= some set).
- L114: Here I was wondering what happens if there are multiple optimal solutions. Maybe you can add some note here. Later in the theorem, you solved it differently.
- L317: "In other words, using the terminology of" sounds a bit redundant.
- L327: I think it must be $X \subseteq V$, because in a graph where all edges are self-loops, we have $X = V$.
- L361: "this problem is not known to be NP-hard" sounds a bit confusing given that it is known to be in $P$.
- L479: "it scales the weight [...] by parameter $\alpha$" Here I was unsure what this means, i.e., if it is $w/\alpha$ or $w \cdot \alpha$. Of course, it is precise in the algorithm.
- L525: "since the Mehlhorn's algorithm is". I think it is "since Mehlhorn's algorithm is".
- L1000: I would have like a bit more details why Theorem 8 implies that Theorem 1 cannot be improved.

---

> ### Author Response · Authors · 2024-11-22
>
> We would like to thank the reviewer for suggesting to add a discussion section and to elaborate more on empirical results, which we have now done, as well as for the many helpful comments on the write-up.
>
> The example in Line 128 can be easily extended to the case where $\eta^- < OPT$:
> Consider an input instance composed of two independent
> parts: Part 1 with optimum value $OPT_1$
> for which we have perfect prediction and Part 2 with optimum value $OPT_2$
> for which we receive the empty prediction, i.e., $\eta^- = OPT_2$.
> Reaching cost smaller than $OPT_1 + \rho\cdot OPT_2$,
> which corresponds to the approximation ratio $1+ (\rho-1)\eta^-/(OPT_1+OPT_2)$
> claimed by our theorem,
> requires finding better than $\rho$-approximate solution on the second part of
> the instance.
> In fact, we have extended Theorem 8 using a similar idea (although it requires a careful argument). Now it reads as follows. There is no learning augmented algorithm for Vertex Cover with performance better than $1+(\eta^+ + \eta^-)/OPT$. This holds for *any* values of $\eta^+/OPT$ and $\eta^-/OPT$ with $\eta^+/OPT + \eta^-/OPT \leq 1$.
>
> Regarding your other comments: we agree with all of them and we have
> addressed them in the uploaded revision of our submission.

---

> > ### Comment · Reviewer_9d1z · 2024-11-25
> >
> > I thank the authors for clarifying my question.
> >
> > I stay with my opinion that the paper is a good addition to the area of learning-augmented algorithms and should be accepted at the conference.

---

### Official Review · Reviewer_iRrN · 2024-11-02

**Soundness:** 3
**Presentation:** 3
**Contribution:** 3
**Rating:** 6
**Confidence:** 4

**Summary:**

The paper explores learning-augmented approximation algorithm design for a general selection problem. In this problem, we are given a set of ground elements, each with a non-negative weight, along with an implicit collection of all feasible subsets of these elements. The objective is to select a feasible subset that minimizes or maximizes the total weight.

The authors focus on the setting where the algorithm can access an (imperfect) prediction of an optimal solution and propose a general framework for integrating classic approximation algorithms with this prediction. For the minimization model, the framework achieves an approximation ratio of $1+(\eta^+ + (\rho-1)\eta^-)/ OPT$, where $(\eta^+,\eta^-)$ are prediction errors, $\rho$ is the classic approximation and $OPT$ is the optimal objective value.  For maximization problems, the framework yields an approximation ratio of $ 1- ((\rho-1)\eta^++\eta^- )/OPT$. The authors apply this framework to several concrete applications. In particular, for the Steiner tree problem, they leverage the characteristics of the problem to provide an improved ratio.

**Strengths:**

- Learning-augmented algorithmic design was originally applied in online optimization to leverage learning in order to address uncertainty. This paper extends this approach to offline approximation algorithm design, aiming to break through computational complexity—an interesting idea.

- The paper is well-organized. The basic idea is clean and easy to follow.

**Weaknesses:**

- Although the model is novel, the proposed learning-augmented framework seems quite natural, and the analysis is technically simple. One shortcoming of the framework is that when the given prediction is the whole element set, $\eta^- =0 $, while $\eta^+$ can become infinitely large, leading to an infinite approximation (if the robust operation in the corollary is not used). This is a little weird. Could this be fixed by adding a step in the framework to apply classic approximation algorithms to the predicted subset of elements if it is infeasible?

**Questions:**

- See the weakness above.

---

> ### Author Response · Authors · 2024-11-22
>
> It is common in previous works on ML-augmented algorithms to allow for predictions with unbounded error, see for instance the papers of Lykouris and Vassilvitskii, or Bamas et al. from the references. This is generally handled via a black-box robustification step (both in the on- and offline setting). For the specific case of the Steiner Tree Problem, we actually show a stronger guarantee on the performance of the algorithm that can be described by a bound which directly depends on the optimum solution. This stronger bound suggests that for the Steiner problem with predictions, robustification may often not be necessary. We actually observe this in our experiments, where we opted not to robustify and still manage to considerably improve over the performance of Mehlhorn's algorithm even with large prediction errors (see Figures 2(a)(e)(f)).
> This improved bound does depend on specific properties of Steiner trees, and we agree that it is an interesting direction for further research to investigate whether similar results are achievable for other problems. We have added this in the newly introduced discussion section of the paper.
>
> We are unsure whether we correctly understand the suggested fix: Note that the cost of an approximation algorithm when run on a subset of the elements can be arbitrarily *higher* than the optimal solution on the whole instance. Thus, in order to estimate the actual optimal value, one has to run the approximation algorithm on the full problem instance (which is precisely what is done in our robustification step).

---

> > ### Comment · Reviewer_iRrN · 2024-11-26
> >
> > Thank the authors for clarifying my concerns.

---

### Official Review · Reviewer_s3yv · 2024-11-03

**Soundness:** 4
**Presentation:** 4
**Contribution:** 3
**Rating:** 8
**Confidence:** 4

**Summary:**

The paper provides a generic transformation of worst-case approximation algorithms to approximation equipped with machine learnt advice. The general setting is that of positively weighted selection problems that are subject to combinatorial constraints (e.g., vertex cover, Steiner tree, weighted machine, knapsack, etc). The main result is that given a nearly correct (yet perhaps unfeasible) solution to the optimization problem, one can derive an approximation guarantee close to 1, using as a black box any approximation algorithm for the problem (in particular, a very efficient one). The idea is very simple. For minimization problems, replace the weights of the elements in the advised solution by 0, then run the approximation algorithm on the modified instance. An analogous solution works for maximization problems.

They also show that under the unique games conjecture, the result is optimal for some problems (e.g., vertex cover). For Steiner tree, they give a better algorithm that is a slight variation of the above general method: instead of zeroing the weight of the advised elements, damp the weights by some factor.

Finally, they provide some empirical evaluation of their methods, on a known benchmark. It's hard to judge the meaning, because it's not clear that this benchmark was design to challenge machine learning approaches, so possibly it is easy to learn and thus overfit the above approach to the dataset (which is quite small; 199 examples).

**Strengths:**

I like the generality of the approach, and its simplicity makes it practically appealing.

**Weaknesses:**

It's restricted to selection problems, so, for instance irrelevant to partition problems such as clustering. Also, it requires an approximation algorithm for the weighted case, even if the optimization problem that needs to be solved is unweighted. By exploring the combinatorial structure of the problem, one might derive better solutions (as demonstrated for Steiner tree).

**Questions:**

None.

---

> ### Author Response · Authors · 2024-11-22
>
> We thank you for your comments.
>
> We agree that although selection problems constitute a broad class of important combinatorial optimization problems, it would be interesting to investigate whether similar results can be obtained for important problems that do not belong to this class.

---

### Official Review · Reviewer_DLoK · 2024-11-03

**Soundness:** 3
**Presentation:** 3
**Contribution:** 3
**Rating:** 8
**Confidence:** 3

**Summary:**

This work falls under the he newly evolving area of learning-augmented algorithms.  Consider an optimization problem of the form:  There are $n$ items with weights $w_1, w_2, \cdost w_n$. Given a (implicit) collection of subsets of $[n]$ find a subset whose weight is minimized/maximized. Suppose that we are given a prediction $\cap{X}$ for the optimal solution. Can the algorithm exploit this additional data and design algorithms with better approximation ratio?

This work studies the above question and proves that depending on how close $\cap{X}$ is to the optimal solution (closeness measured in terms of false positives and false negatives), we can obtain algorithms with improved approximation ratio.

**Strengths:**

1. The problem studied is very interesting, natural, and timely
2. The solution presented is very intuitive and the proofs are easy to follow (this should be taken as negative).
3. For the general setting, the bounds obtained are optimal.

**Weaknesses:**

1. The paper is a bit too verbose with many unnecessary details.   For example, detailed discussed of example applications in sections 2.1 and 3.1 is not needed, as these problems fit into the framework in a straightforward manner.

2. I do not know much about the learning-augmented algorithms. So, I am not able evaluate the novelty of the proofs of the present work (this perhaps is the reviewer's weakness, not the paper's weakness). However, a more detailed description of various other models to represent predictions and the algorithmic techniques places this work in context.

**Questions:**

1. What if predictions come in the form of probabilities/confidences? For each item  $i$, a confidence value $\alpha_i$, representing the confidence that item $i$ is in the optimal solution set.  How does this model compare to the current work?

---

> ### Author Response · Authors · 2024-11-22
>
> The suggested model where predictions come in the form of probabilities/confidences is actually captured by our model: one can round such a "partial" prediction to a "real" prediction simply by selecting each item to be part of the prediction with probability proportional to the corresponding probability/confidence score. By linearity of expectation, $\eta^+$ and $\eta^-$ are preserved and one can apply our framework.
> An alternative approach would be to set $\bar{w}(i)$ to $(1–\alpha_i) \cdot w(i)$ (as a generalization of setting it to $0$ iff $i \in \hat{X}$) and keep the rest of the algorithm identical. This latter approach might perhaps be more natural and easier to apply in practice, but would lead to a significantly more involved analysis. Thank you for raising this important point, we have added a comment about this in the paper.
>
> Regarding your question on different models to represent predictions, and the respective algorithmic techniques: as mentioned in the paper, much of the to-date research on learning augmented algorithms focuses on online problems. In contrast, offline problems were mostly studied in the warm-start setting, where predictions, typically coming from solutions to past instances, are used to speed up exact algorithms. The challenges of each setting are different: For online problems, predictions are typically used to reduce the uncertainty about the future parts of the input. Here, the main obstacle is devising a robust algorithm that incorporates the predictions, while ensuring feasibility is (in general) easy.
> For offline problems, the challenges are different. So far, the focus has been on the dependence between the running time required to get an optimal solution and the quality of the prediction provided. In contrast, we maintain a superb running time in all situations and study the dependence between the approximation ratio and the quality of the prediction. While the L1 norm is a very popular choice for the prediction error across many different settings (online and offline), the techniques can differ significantly in each of them. In particular, we are not aware of any work that uses similar techniques to ours.
>
> Finally, we appreciate the feedback on our writing style, but still find the discussion of example applications useful in order to showcase how our algorithm situates between prior (classic) results; moreover, some of the applications (Matching, Knapsack) require small but nontrivial arguments which we prefer to provide explicitly in the paper.

---

> ### Comment · Reviewer_DLoK · 2024-11-25
>
> Thank you for the detailed response addressing the question and incorporating the discussion about confidence/probability. I revising my score.

---

### Meta-Review · Area_Chair_ZZi9 · 2024-12-21

**Metareview:**

The paper designs learning augmented algorithms for a broad class of combinatorial optimization problems. The main contribution of this work is a general framework for deriving learning augmented algorithms from worst-case approximation algorithms. The framework applies to many classical optimization problems where the goal is to select a feasible set of items with maximum or minimum weight. The paper shows that the results are optimal for certain problems.

The reviewers appreciated the theoretical contributions of this work. The reviewers agreed that the contribution is strong and it is a valuable addition to the area of learning-augmented algorithms. The main weaknesses raised by the reviewers were that the framework is limited to selection problems, and the approach is very simple both conceptually and technically. Nevertheless, selection problems are a broad class and the theoretical contributions are strong.

**Additional Comments On Reviewer Discussion:**

The reviewers asked several clarifying questions that were addressed by the authors. After the discussion with the authors, there was strong consensus among the reviewers that the paper makes a valuable contribution to the area and it should be accepted.

---

### Decision · Program_Chairs · 2025-01-22

Accept (Spotlight)